# Global land and water limits to electrolytic hydrogen production using wind and solar resources

Davide Tonelli ©[1,2,3] ✉, Lorenzo Rosa ©[3] ✉, Paolo Gabrielli ©[3,4], Ken Caldeira ©[3,5], Alessandro Parente[2] & Francesco Contino[1]

Proposals for achieving net-zero emissions by 2050 include scaling-up electrolytic hydrogen production, however, this poses technical, economic, and environmental challenges. One such challenge is for policymakers to ensure a sustainable future for the environment including freshwater and land resources while facilitating low-carbon hydrogen production using renewable wind and solar energy. We establish a country-by-country reference scenario for hydrogen demand in 2050 and compare it with land and water availability. Our analysis highlights countries that will be constrained by domestic natural resources to achieve electrolytic hydrogen self-sufficiency in a net-zero target. Depending on land allocation for the installation of solar panels or wind turbines, less than 50% of hydrogen demand in 2050 could be met through a local production without land or water scarcity. Our findings identify potential importers and exporters of hydrogen or, conversely, exporters or importers of industries that would rely on electrolytic hydrogen. The abundance of land and water resources in Southern and Central-East Africa, West Africa, South America, Canada, and Australia make these countries potential leaders in hydrogen export.

Electrolytic production of hydrogen using low-carbon electricity can contribute[1–3] to achieve net-zero greenhouse gas (GHG) emission goals and keep global warming below 2 °C. In 2020, global hydrogen production reached 94 million tons per year (Mt/y) and is expected to rise[4] to 530 Mt/y by 2050. Presently, 98% of hydrogen production is based on fossil feedstocks[5]. Steam methane reforming, using natural gas, accounts for 75% of global production, followed by coal gasification (23%) and oil/naphtha reforming (<0.1%)[5]. Alternative low-carbon methods for hydrogen production involve coupling steam methane reforming with carbon capture (blue hydrogen) or utilizing renewable natural gas from biomass through anaerobic digestion[6,7]. Methane pyrolysis enables simultaneous hydrogen and carbon production, with carbon sequestration in a solid form[8,9]. However, these technologies face limitations, such as the local availability of sustainable biomass feedstocks[6], infrastructure requirements for carbon capture, transport, and storage[10,11], methane leakage management[12], and limited $CO_2$ capture rates at the plant level (55–93%)[7].

Natural gas is a potent greenhouse gas (GHG) with a global warming potential 28 times higher than $CO_2$ over a 100-year timescale[13]. The leakage rate in the production and transportation system of natural gas is a relevant factor affecting the carbon intensity of blue hydrogen[14–16]. In contrast, hydrogen produced through water electrolysis has a carbon intensity linked to the source of electricity and, to a lesser extent, the manufacturing of the system[17]. Currently, water electrolysis accounts for only 2% of global hydrogen production[5]. However, its widespread adoption alongside renewable

[1]Institute of Mechanics, Materials and Civil Engineering, UCLouvain, 1348 Ottignies-Louvain-la-Neuve, Belgium. [2]Aero-Thermo-Mechanics Department, ULB, 1050 Brussels, Belgium. [3]Department of Global Ecology, Carnegie Institution for Science, Stanford, CA 94305, USA. [4]Institute of Energy and Process Engineering, ETH Zurich, 8092 Zurich, Switzerland. [5]Breakthrough Energy, Kirkland, WA 98033, USA. ✉e-mail: davide.tonelli@uclouvain.be; lrosa@carnegiescience.edu

technologies like solar and wind power could facilitate large-scale production of low-carbon hydrogen.

Large-scale deployment of electrolytic hydrogen raises concerns about the availability of sufficient land and water resources for the installation of solar photovoltaic panels, wind turbines, and water electrolysis systems. Existing analyses[18–21] consider only global-scale assessments of land and water availability, independent of country-specific limits, when examining the production and trade of hydrogen from countries with abundant renewable resources to those with limited renewable energy potential. Dedicating land to the installation of renewable technologies can reduce the availability of land for cropland expansion, hence limiting the amount of land dedicated to food production. The competition in land uses, combined with the rise in population, lead to a decrease in the per capita availability of arable land[22]. The growing production of biofuels for energy also contributes to increased competition for land between energy and food systems[23,24]. In addition, the infrastructure required for wind and photovoltaic energy can have considerable direct environmental impacts, including habitat loss and biodiversity decline[25]. These impacts can extend beyond the occupied land area, affecting animal species and ecosystem responses[26].

Global demand for water has been steadily increasing at a rate of 1% per year[27], accompanied by a progressive decline in available water resources and an increase in water pollution[28]. These trends can be attributed to population growth, climate change, and economic development. The uneven distribution of water availability, coupled with varying economic growth rates among countries, has intensified water scarcity and led to competition in water usage among energy, industrial, and municipal systems[29]. Total water demand for hydrogen production is projected to have relatively minimal hydrological implications on a global scale compared to water demand for food production[18]. Among the end-uses, the estimated water demand for global electrolytic hydrogen in a net-zero emissions chemical industry is <1.5% of the water demand for food production[30]. However, local and regional water constraints have been recognized among the key factors that define a country's capability to produce and export electrolytic hydrogen. At the country or local level, the concentration of plants manufacturing photovoltaic panels and wind turbines, or factories operating electrolyzers, in areas affected by water scarcity can exacerbate the limitations in water availability due to competing demands for water resources. These constraints are considered alongside the availability of renewable resources and infrastructure development[19,31]. It is worth noting that regions with abundant solar resources also tend to face water scarcity, and it is anticipated that the majority of planned electrolyzer capacity will be located in such regions[20,32].

In this work, we focus on assessing the global demand and availability of land and water resources at the country level for prospective large-scale electrolytic hydrogen production using wind and solar electricity. This study builds upon previous literature[4,5] by providing reference cases that disaggregate hydrogen demand by sector and by country for both 2020 and 2050. These findings can serve as inputs for future techno-economic analyses. Moreover, we identify the potential for electrolytic hydrogen production by combining spatially-explicit data (i.e., high-resolution rasterized geospatial data) on the power production yield of solar and wind technologies with country-specific land availability[33,34]. The analysis of land and water requirements, based on different levels of renewable technology penetration, highlights countries that will be constrained by domestic natural resources to achieve electrolytic hydrogen self-sufficiency in a net-zero target in our reference scenarios. Consequently, by identifying countries that could become future importers or exporters of electrolytic hydrogen (or, conversely, exporters or importers of industries that would rely on electrolytic hydrogen), this work contributes to the discussion on the geopolitical implications of a large-scale hydrogen economy[19,20].

## Results
### Country-specific hydrogen demand

We estimate the potential demand for hydrogen at both country and sector levels for years 2020 and 2050. Detailed country- and sector-specific results can be found in the Supplementary Dataset. While the hydrogen demand in 2020 is currently met by fossil resources, our 2020 scenario represents a counterfactual situation where all fuel or feedstock inputs to the chemical, cement, refineries and light industry, steel production, and transport sectors are instead provided by electrolytic hydrogen (see Methods subsection "Hydrogen demand").

Figure 1 illustrates the electricity demand necessary for electrolytic hydrogen production across different sectors. The left y-axis represents the per capita capacity required for electrolytic hydrogen production, while the right y-axis compares the per capita capacity for electrolytic hydrogen with the per capita capacity for direct electricity consumption. There is a substantial difference between 2020 and 2050, primarily driven by an 80% reduction in hydrogen demand in refineries, a threefold increase in hydrogen demand in the chemical sector (ammonia and methanol production), and the transport sector transitioning from zero to over 200 Mt/y globally.

In our reference scenario, all countries depicted in Fig. 1 experience a larger increase in per capita hydrogen demand from 2020 to 2050 (on average seven times higher demand in 2050 than in 2020) compared to the variation in the per capita direct electricity demand (on average two times higher in 2050 than in 2020). This trend is particularly noticeable in the United States, Brazil, and Mexico. However, exceptions include Malaysia, South Korea, and the Netherlands, where the rise in hydrogen demand is similar or lower than the increase in electricity demand. Specifically, in 2020, the United States would have required a capacity of 0.2 kW per person for hydrogen production if electrolytic hydrogen was used. This capacity would subsequently increase to 1.2 kW per person by 2050. This corresponds to 15% and 70% of the capacity needed for direct electricity demand, respectively. Trinidad and Tobago, known for its extensive fossil fuel industry, stands out as an outlier with required capacity for electrolytic hydrogen production in our reference cases being 14 to 17 times higher than the capacity for direct electricity demand in 2020 and 2050, respectively.

### Land requirements

Figure 2a, b presents the per capita land requirements to meet hydrogen demand in our reference cases and compares it with the eligible land (as defined in Methods subsection "Land scarcity") in selected countries for 2020 and 2050. The eligible land of each country is determined by subtracting the fractions of the area occupied by agriculture, artificial surfaces, and forests (which are considered ineligible for the installation of renewable energy systems) from the total area of the country. This calculation helps identify the portion of land that can be utilized for the installation of renewable energy infrastructure, such as solar panels and wind turbines, to meet energy demands sustainably. The interval bars represent the range between two extreme scenarios of hydrogen demand in 2050 (92 Mt/y and 646 Mt/y globally), while the country-specific land requirement corresponds to the reference scenario of 400 Mt/y of hydrogen demand globally. Depending on the scenario of hydrogen demand, land requirements for 2050 hydrogen demand vary between 0.09 and 0.6 million km$^2$ for solar panels and 1.9 and 13.5 million km$^2$ for wind turbines (Fig. 2c, d). Country-specific land use is calculated by dividing the electricity needed for hydrogen production by the electricity produced per unit area from solar panels and wind turbines (Fig. 2a, b). Theoretical coverage of 100% of eligible land with solar panels or wind turbines ($f^{coverage} = 100\%$) is shown. However, practical constraints such as economic, environmental, and socio-political factors limit this potential[35]. Gray lines in Fig. 2a, b represent different fractions of eligible land coverage ($f^{coverage}$) that are compared to the land requirements for hydrogen presented in Fig. 1. Trinidad and Tobago (not

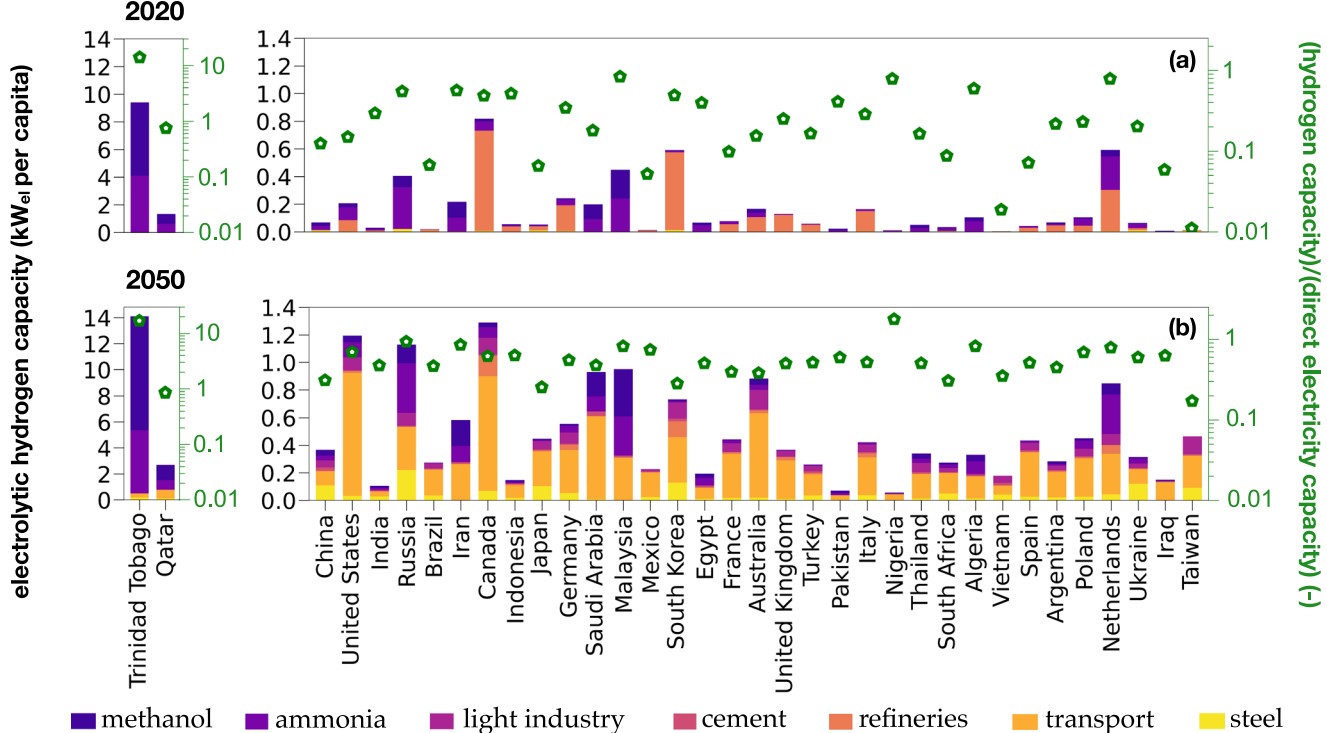

**Fig. 1 | Country-specific hydrogen demand in 2020 and 2050.** Hydrogen demand and capacity comparison for selected countries in **a** 2020 and **b** 2050. The figure illustrates the hydrogen demand per sector for the top 35 countries ranked by country-specific demand in 2050. The stacked bars on the left-hand side y-axis represent the per capita capacity required for electrolytic hydrogen production in each sector. The green markers on the right-hand side y-axis represent the ratio of the capacity required for electrolytic hydrogen production to the capacity required for direct electricity demand. To derive the capacity for direct electricity demand (kW_el per capita), divide the capacity for electrolytic hydrogen (y-axis, left-hand side) by the corresponding ratio of the capacity for electrolytic hydrogen to direct electricity demand (y-axis, right-hand side). The conversion factors used are: $1 \, kW_{el} = 0.62 \, kW_{H2}$, $1 \, kWh_{H2} = 0.03 \, kg_{H2}$, and $1 \, kW_{el} = 8760 \, kWh_{el}/y$. The countries are ordered based on their total hydrogen demand in 2050, apart from Trinidad and Tobago and Qatar. Note: the Supplementary Dataset provides country-specific results for hydrogen demand and capacity.

shown in Fig. 2 for scaling reasons) has sufficient available land for hydrogen production only when considering electricity production from solar panels and the hydrogen demand of 2020. In this case, 80% of eligible land coverage would be required. Conversely, several countries, including Canada, Australia, Russia, Algeria, and Argentina, would only need to cover 1% of their eligible land with solar panels to meet their projected hydrogen demand in 2050 (Fig. 2a). However, when wind turbines are considered, the land requirements can be more than ten times larger than that of solar panels. In this scenario, countries such as South Korea and Japan, which have limited land availability compared to per capita hydrogen demand, require more land than what is eligible (Fig. 2b). Variations in future hydrogen demand due to different demand scenarios for 2050 do not affect the presence or absence of water scarcity, except for the Netherlands and South Korea in case of power production from wind turbines. In these countries, contrary to the reference demand scenario, the lower demand scenario would not result in land scarcity.

**Water requirements**

Figure 3a, b illustrates the country-specific amount of water required to meet electrolytic hydrogen demand and other sectors' demand in our reference cases. The interval bars represent the range between two extreme scenarios of hydrogen demand (92 Mt/y and 646 Mt/y globally), while the country-specific water requirements correspond to the reference scenario of 400 Mt/y of hydrogen demand globally. Depending on the scenario of hydrogen demand, water requirements for 2050 hydrogen demand vary between 13.6 and 95.6 billion m³ in the case of solar panels and 3.2 and 22.6 billion m³ in case of wind turbines. Compared to the combined global water withdrawals

for agriculture, industry, and municipalities (3976 billion m³)[36], the water required for hydrogen production accounts for <3% (Fig. 3c, d). Figure 3a, b also compares these withdrawals with available renewable water resources in selected countries, as defined in Methods (see "Water scarcity"−Eq. 8), for the years 2020 and 2050. The height of the bars represents the numerator in the calculation of water scarcity per country (see Methods subsection "Water scarcity"−Eq. 10), while the contours in the background represent the denominator. The light gray line in Fig. 3 corresponds to the 100% utilization of water available, which represents the sustainability limit for water exploitation. This value is derived by subtracting the environmental flow requirements− the quantity, timing, and quality of freshwater flows required to sustain freshwater and estuarine ecosystems−from a country's available renewable water resources (see Methods subsection "Water scarcity"− Eq. 8)[37–39]. In contrast to land use (Fig. 2), the manufacturing and operation of wind turbines require a smaller amount of water compared to solar panels (values reported in Suppl. Inf. - S.4.1 Water requirements). Independently of the electricity generation technology, several countries, which include Saudi Arabia, Algeria, Trinidad and Tobago, China, India, Egypt, Turkey, and South Korea, already exceed their sustainable domestic water resources. When comparing water requirements for hydrogen production to water withdrawals in agriculture, industry, and municipal uses, hydrogen's water requirements account for <5% of the total water withdrawals in each country. In most countries, the water required for hydrogen is less than water withdrawals for other uses such as agriculture, municipality, and industry. However, in some countries like Trinidad and Tobago (as well as Iceland, Luxembourg, and Qatar not appearing in Fig. 3), the water needed for hydrogen production can be more than one order of

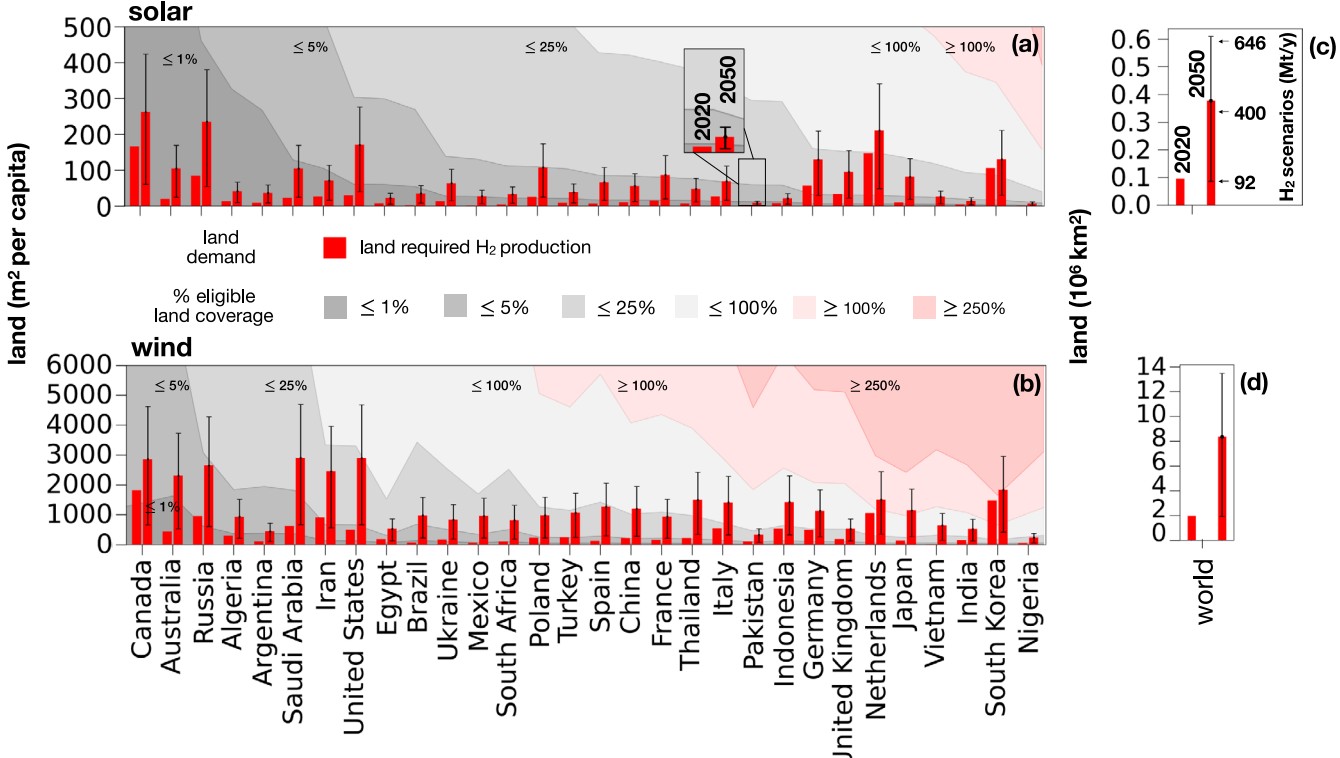

**Fig. 2 | Land requirements to fulfill hydrogen demand in 2020 and 2050.** Per capita country-specific land requirements to fulfill hydrogen demand in 2020 and 2050: **a** solar and **b** onshore wind power production. The figure shows the land requirements to fulfill hydrogen demand **a, b** at country level as per capita land requirements and **c, d** in terms of total global land requirements in 2020 and 2050 based on **a, c** solar and **b, d** onshore wind power production. The countries are selected from the top 30 in terms of total demand of hydrogen in 2050 and are ordered according to the amount of eligible land for solar panels (excluding Trinidad and Tobago for scaling reasons). The interval bars represent the range of land required in 2050 based on the reference and the extreme scenarios of global hydrogen demand in 2050: 400 Mt/y (reference), 92 Mt/y (smallest), 646 Mt/y (largest). The gray areas (and pink areas) represent fractions (and multiples) of the eligible land, as defined in Methods subsection "Land scarcity", and are used for comparison with the land requirements for hydrogen demand in each country. Different energy generation technologies require varying amounts of land for the same amount of hydrogen demand.

magnitude greater (in the case of solar panels) than the water withdrawals in agriculture, municipality, or industry. In Trinidad and Tobago, hydrogen demand is the primary contributor to water scarcity, whereas in other countries, hydrogen demand exacerbates existing water scarcity conditions. Except in Trinidad and Tobago, where hydrogen demand is a main driver of water requirements, variations in hydrogen production due to different demand scenarios for 2050 do not create water scarcity in any other country.

## Land scarcity
Figure 4 depicts the extent of land scarcity across all countries, considering varying fractions of eligible land coverage ($f^{coverage}$) for the installation of solar and onshore wind technologies in our reference cases, as defined in Methods subsection "Land scarcity". The eligible land of a country refers to the total land area that can be technically covered with renewable technologies ($f^{coverage} = 100\%$). However, additional economic and socio-political constraints limit the practical coverage of land with renewable technologies, which is quantified by the fraction of eligible land coverage ($f^{coverage} < 100\%$). By comparing the land required for electrolytic hydrogen production with the land practically covered by renewable technologies (LS in Methods subsection "Land scarcity"—Eq. 7), we assess the presence (LS > 1) or the absence (LS < 1) of land scarcity resulting from hydrogen production. When assuming 100% land coverage for solar panels installation ($f^{coverage} = 100\%$), only Trinidad and Tobago experiences land scarcity (Fig. 4a). However, other countries with high electrolytic hydrogen demand in our reference cases and limited land availability face land scarcity when producing hydrogen using electricity from wind

turbines. This includes Belgium and the Netherlands in Europe, as well as Equatorial Guinea, Japan, South Korea, and Malaysia in Asia (Fig. 4b). Similarly, land scarcity arises when installing solar panels in scenarios with eligible land coverage ($f^{coverage}$) below 10%. In such cases, countries with lower population density, but considerable industrial activity, like Germany, the United Kingdom and Austria, also encounter scarcity (Fig. 4c, e). Notably, considering the same fraction of land coverage below 10%, a greater number of countries experience land scarcity when installing wind turbines alone (Fig. 4d, f).

## Water Scarcity
Figure 5 highlights the exacerbation in water scarcity compared to the scenario without hydrogen production (see Methods subsection "Water scarcity"—Eqs. 9, 10). Except for Trinidad and Tobago, where hydrogen production from solar panels can lead to water scarcity, the assumed reference scenario for hydrogen demand does not create water scarcity anywhere in the world if water scarcity is not already present (the relative increase in water withdrawals due to hydrogen production is shown in Suppl. Inf. - Section S.4.2 Impact of water requirements for hydrogen production). This is because water withdrawals for hydrogen production are negligible compared to total water withdrawals in other sectors (agriculture, industry, and municipal demand) (see Suppl. Inf. - S.4 Water demand). However, in 37 countries already facing water scarcity, hydrogen production exacerbates water scarcity. The most affected countries are in the North African and Middle East regions (MENA). Our analysis reveals that 5 countries experience an increase in water scarcity >5% when using solar panels, while 16 countries experience a relatively minor exacerbation below 1%

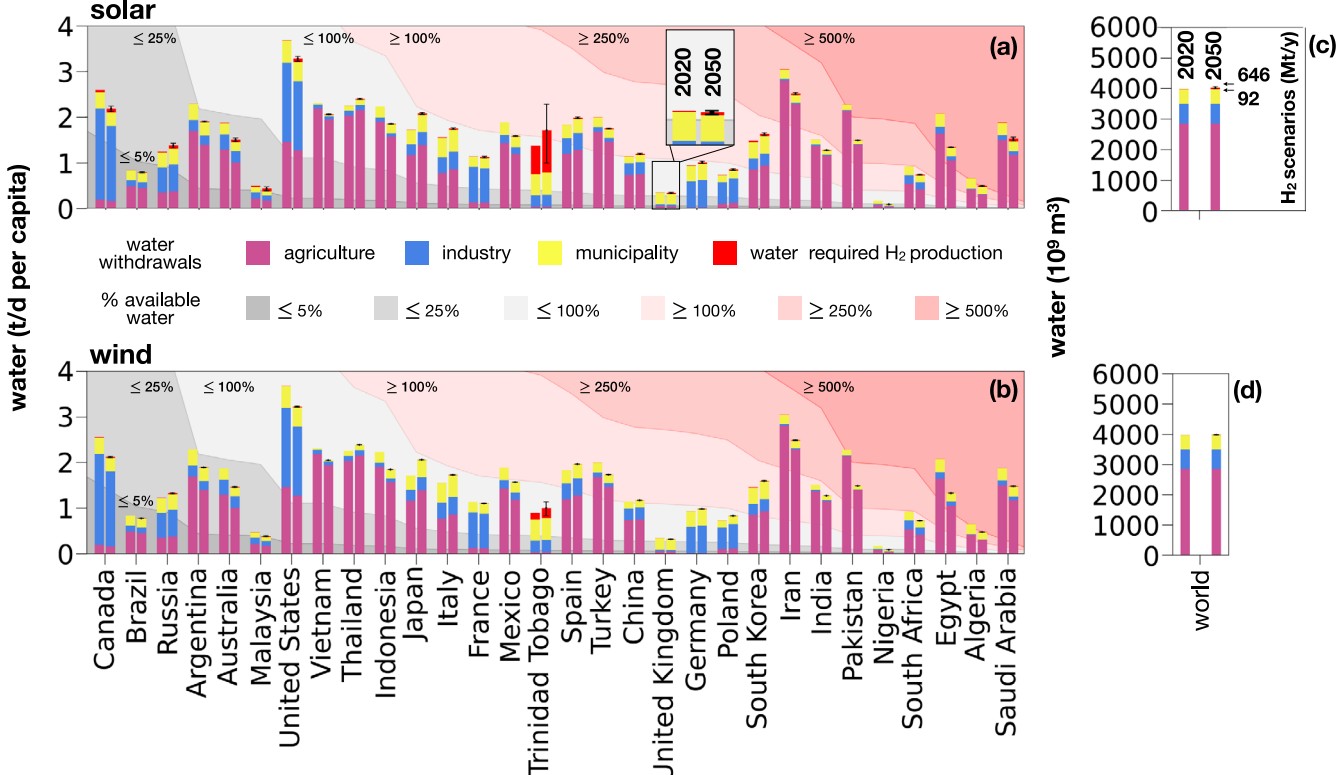

**Fig. 3 | Water requirements to fulfill hydrogen demand in 2020 and 2050.** Per capita country-specific water requirements to fulfill hydrogen demand in 2020 and 2050: **a** solar and **b** onshore wind power production. The figure shows the water requirements to fulfill hydrogen demand **a**, **b** at country level as per capita water requirements and **c**, **d** in terms of total global water requirements in 2020 and 2050 based on **a**, **c** solar and **b**, **d** onshore wind power production. The countries are selected from the top 30 in terms of total hydrogen demand in 2050 and are ordered based on the amount of renewable available water. The interval bars represent the range of land required in 2050 based on the reference and the extreme scenarios of global hydrogen demand in 2050: 400 Mt/y (reference), 92 Mt/y (smallest), 646 Mt/y (largest). The gray areas represent the fractions of available water resources (water resources minus environmental flow requirements) in each country, corresponding to the water withdrawals. The red areas indicate water withdrawals exceeding the water availability limit (100%, 250%, and 500%). The purple, blue, and yellow stacks represent water withdrawals in agriculture, industry, and municipality, respectively. The red stacks represent the water required for hydrogen demand in 2020 and 2050. Total water withdrawals exceeding 100% indicate the presence of water scarcity.

(Fig. 5a). Specifically, Oman and Qatar demonstrate an increase in water scarcity >5% when using solar panels (Fig. 5a), and Qatar experiences this increase when utilizing wind technologies (Fig. 5b).

## Discussion

The predictions for hydrogen demand in 2050 vary substantially across scenarios, ranging from 3% to 12% of the global total final energy consumption, corresponding to a range of 92 Mt/y to 646 Mt/y[40]. In our reference scenario with hydrogen demand of 400 Mt/y in 2050, we find that with a threshold of 5% eligible land coverage ($f^{coverage} \leq 5\%$) using electricity production from solar panels, 10 Mt/y of demand is located in countries constrained by both land and water resources (3%), 43 Mt/y constrained by land resources (11%), and 163 Mt/y constrained by water resources alone (41%). Furthermore, 184 Mt/y of hydrogen demand (46%) can be met with local production where neither land nor water scarcity is present. Figure 6 combines the analysis of land and water scarcity, highlighting countries constrained by both land and water resources (red), land resources alone (yellow), water resources alone (blue), and those with no land and water limitations (green). These findings are highly dependent on the energy generation technology and the assumed fraction of eligible land coverage ($f^{coverage}$) (see sensitivity analysis in Suppl. Inf. - Section S.9 Sensitivity number of countries with land and water scarcity). Water scarcity is predominantly observed in the MENA region, South Africa, India, China, and certain countries in Central Asia. Among the countries facing both land and water scarcity (red countries in Fig. 6a, b),

Trinidad and Tobago experiences it when considering 100% eligible land ($f^{coverage} = 100\%$) and full water resource exploitation in the case of solar panels, while Belgium and South Korea experience it in the case of wind power. The comparison of results for various demand scenarios is summarized in Suppl. Inf. - Section S.9 Sensitivity number of countries with land and water scarcity.

In contrast, when considering only 5% of eligible land for the installation of solar panels ($f^{coverage} = 5\%$) and hydrogen demand from our reference cases, Belgium, Trinidad and Tobago, South Korea, and Dominican Republic are the countries that experience both land and water scarcity (red countries in Fig. 6e). However, if we assume renewable electricity solely generated from onshore wind, the number of countries facing both land and water scarcity increases to 20, including Spain among the European countries, South Africa, Middle East countries, India and China (red countries in Fig. 6f). Land scarcity alone emerges in European countries under more restrictive scenarios of eligible land coverage with renewable technologies. The disparity between using solar and wind energy is substantial. With 10% eligible land coverage ($f^{coverage} = 10\%$), three countries experience contemporary land and water scarcity for solar (Fig. 6c) while the number rises to 14 countries for wind (Fig. 6d). Under the assumption of 5% eligible land coverage ($f^{coverage} = 5\%$), 45 and 94 countries encounter land or water scarcity, or both, when utilizing solar (Fig. 6e) and wind energy (Fig. 6f), respectively. It should be noted that land scarcity in countries with large geographical extensions, such as the United States, Brazil, Russia and China, arises due to a relevant portion of land being covered by forests

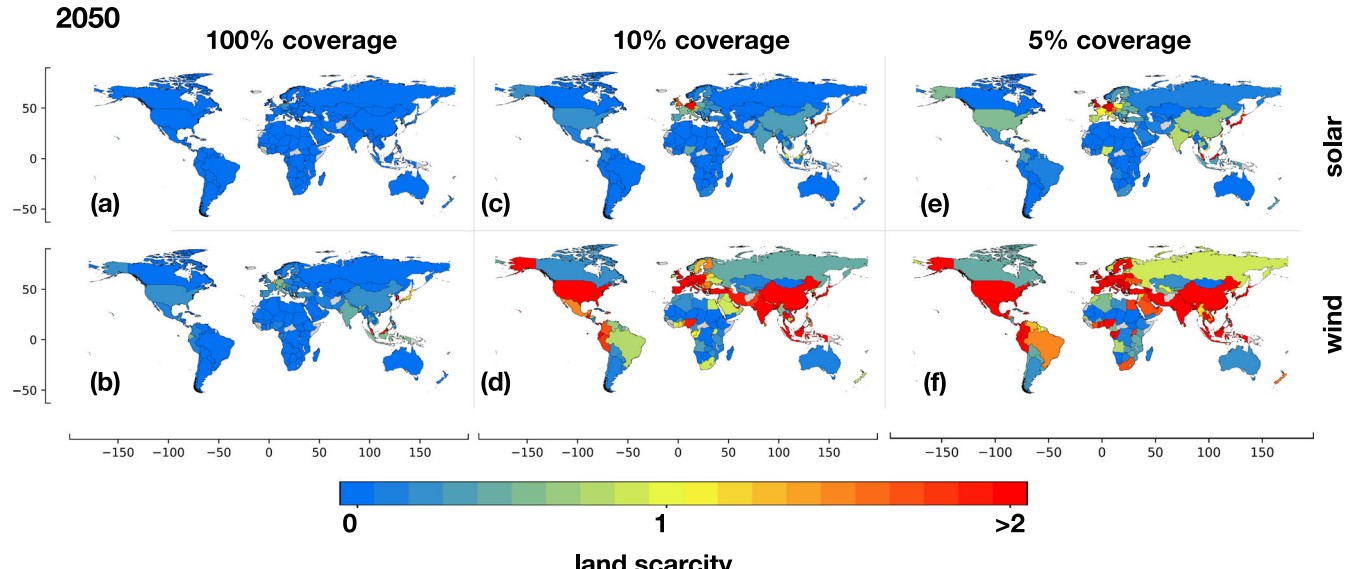

**Fig. 4 | Land scarcity induced by hydrogen production in 2050.** Land scarcity induced by hydrogen production in 2050 across countries worldwide, considering various fractions of land coverage for **a**, **c**, **e** solar and **b**, **d**, **f** onshore wind power production. The degree of land scarcity depends on the assumed eligible land coverage ($f^{coverage}$) for renewable technologies in each country, reflecting economic and socio-political constraints. Notably, the lower power density of onshore wind leads to a significantly higher number of countries experiencing scarcity. Countries depicted in gray lacked available data. The maps are created with the Matplotlib and Geopandas packages for Python[101,102].

and/or agriculture. Countries that experience no scarcity under any condition include Australia, Canada, Argentina, Mongolia, Namibia, and a substantial number of countries in Central Africa.

Land and water scarcity pose challenges for several countries, including Western European countries, Japan, the Dominican Republic, Trinidad and Tobago and South Korea, even when assuming the most efficient use of available land for solar panels installation. These countries, in their pursuit of achieving the net-zero target, may face the need to relocate industrial production plants to regions with greater renewable energy availability, therefore offshoring industrial activities. Alternatively, if there is no considerable reduction in their electrolytic hydrogen demand by 2050, these countries may have to rely on importing electrolytic hydrogen from other nations. Conversely, countries in Southern and Central-East Africa (from Namibia to Eritrea), West Africa (from Eritrea to Mauritania), South America (Argentina, Bolivia and Paraguay), as well as Canada and Australia, are not affected by land or water scarcity, regardless of the power production technology employed. These countries could, therefore, be well-positioned to become major exporters of hydrogen. It is important to note that while land and water availability play crucial roles, other factors such as infrastructure development for hydrogen production and transportation also define a country's potential to become exporter of electrolytic hydrogen[19]. Countries with the largest surplus or deficit of electricity for electrolytic hydrogen production are summarized in Suppl. Inf. - Section S.11 Potential hydrogen exporters and importers.

When considering energy-intensive industries such as steel, cement, ammonia, and methanol, we made the decision to allocate hydrogen demand based on the existing production plants rather than the potential future locations of final products consumption. This approach is due to the uncertainties surrounding the future locations, which depend on various factors. For energy-intensive industries, the cost of energy is an important component in the overall cost of final products[20]. Historically, production plant locations for these industries have been driven by access to inexpensive fossil resources[41]. However, the implementation of carbon emissions reduction initiatives has encouraged the adoption of direct or indirect electrification methods, including electrolytic hydrogen, to replace fossil-based processes. Access to affordable local renewable energy has become a crucial

factor influencing the relocation of these industries. Other factors contributing to relocation decisions include material, labor, and capital costs, which are influenced by country-specific policies and corporate strategies. Future consumption patterns in the chemical and petrochemical industry are influenced by market dynamics and the adoption of new end-use technologies. Due to the numerous factors that can affect these industries, we are cautious about making assumptions regarding their displacement. Therefore, our analysis in this study is based on historical data from 2020. Trinidad and Tobago serves as an exemplary case where the country's wealth is substantially dependent on the energy-intensive chemical industry, which relies on local natural gas resources. However, emissions limitations, the potential depletion of fossil resources within the country, and the limited availability of renewable resources expose the country to the risk of deindustrialization or dependence on low-carbon hydrogen imports from neighboring countries.

In our findings, Trinidad and Tobago appears as an outlier in terms of the impact of hydrogen demand on land and water requirements. Despite having an estimated population of only around 1 million people, Trinidad and Tobago is one of the world's largest exporters of ammonia and methanol. The country's ability to maintain its leading position as an exporter of chemical products in a net-zero economy hinges on its capacity to achieve a production level of 165 TWh/year of low-carbon hydrogen by 2050, as per our reference scenario. Our results underscore that meeting this required production capacity of low-carbon hydrogen necessitates a combined and highly intensified approach involving the installation of solar panels, onshore, and offshore wind power technologies for electrolytic hydrogen production. The country can also leverage its expertise from the oil and gas industry and its advantageous access to depleted oil and gas fields to continue harnessing its gas deposits for low-carbon hydrogen production, utilizing carbon capture and storage technologies[6,7]. In other countries with high per capita hydrogen demand and limited renewable resources, such as Japan, South Korea, and various European countries, land scarcity arises from the limitation in eligible land for the installation of solar panels and wind turbines. Among these countries, Japan is planning to import 0.3 million tons per year (Mt/y) of hydrogen, establishing an international supply chain through

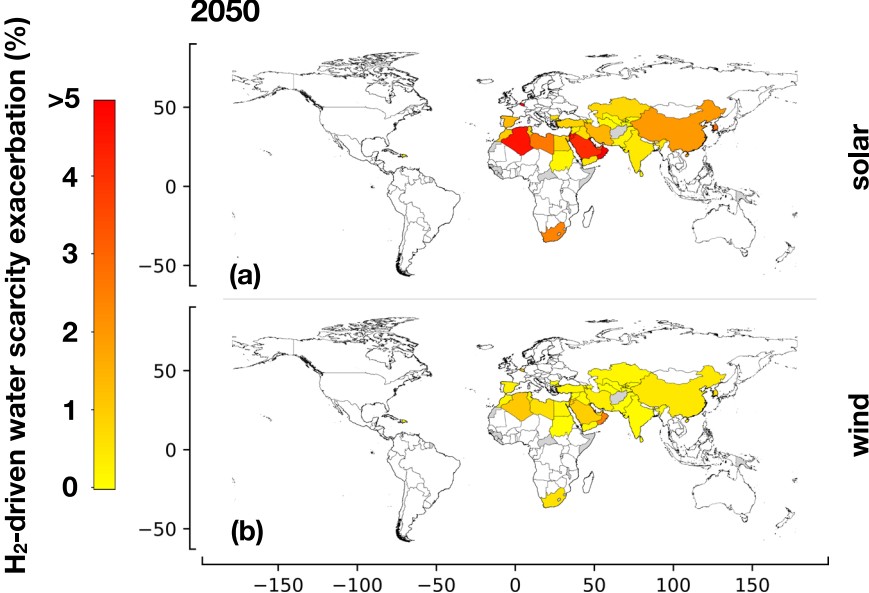

**Fig. 5 | Exacerbation of water scarcity induced by hydrogen production in 2050.** Exacerbation of water scarcity resulting from additional water required for hydrogen production in 2050 through water electrolysis, compared to current water withdrawals for agricultural, industrial, and municipal activities. Power production from **a** solar panels and **b** onshore wind. The demand for water in hydrogen production alone does not create water scarcity in countries where it is not already present. However, the additional water demand for hydrogen production can exacerbate scarcity in countries already affected by water scarcity. Countries without color in the figure do not experience water scarcity. Gray-colored countries indicate unavailable data. The maps are created with the Matplotlib and Geopandas packages for Python[101,102].

collaborations with Australia and Brunei[42]. In addition, the European Union has set a target of importing 50% of its hydrogen supply by 2030[43], with Belgium specifically aiming to import 100% of its hydrogen supply[44] (see Suppl. Inf. - S.1.3 Country-specific hydrogen strategies). The assumptions made in this study consider current values of water and land availability, which are influenced by existing geographical and hydrogeological conditions. It is important to acknowledge that these conditions have the potential to undergo important changes over the 27-year period from the present time. An additional assumption of this study is the production of solar panels and wind turbines to occur within the same country as their installation sites. While this assumption simplifies the analysis, it does not fully capture the complexities of global supply chains behind these technologies. However, one of the main findings of this work is the relatively minor impact of water demand for low-carbon hydrogen production in the creation of water scarcity (with the exception of Trinidad and Tobago). Accordingly, we do not expect that incorporating a model of the supply chain for photovoltaic panels and wind turbine supply would materially alter the conclusions drawn from our work.

Our findings support the conclusion that, on a global scale, the water demand for hydrogen production is negligible compared to the 10,560 billion m³ of water globally available[36] (renewable water resources minus environmental flow requirements). The water demand for hydrogen production represents a small fraction, ranging between 0.13% and 0.56% of the total water available in our analysis (see Suppl. Inf. - S.4 Water demand). However, it is important to note that water availability and demand for electrolytic hydrogen are not evenly distributed across countries. Consequently, not all countries possess equal suitability for hydrogen production. Therefore, local water assessments are crucial to determine the feasibility of each hydrogen production project. While solar panels generally have a higher power yield per unit of area covered compared to wind turbines, the manufacturing process of the former typically requires a larger water footprint than the latter. Water requirements in the manufacturing and operation of solar panels primarily stem from silicon production in photovoltaic systems (60%), cell manufacturing (10%), and module assembly (10%)[45]. For onshore wind turbines, water requirements are mainly associated with the manufacturing of the tower (40%), blades (25%), and generator (20%)[45]. In the case of electrolysis, water requirements consist of the stoichiometric water demand (9 $l_{H_2O}/kg_{H_2}$–38% of the total water demand) and additional water demand for the water treatment process (15 $l_{H_2O}/kg_{H_2}$–62% of the total water demand)[46]. It should be noted that this study does not consider water demand for the cooling process of the electrolyzer, which could potentially increase the water demand by an additional 19%[46]. Regarding land demand in photovoltaic and wind turbine systems, it can be mitigated through design optimization, the use of innovative materials, and careful site selection. Furthermore, solar panels can be installed in agrivoltaic systems, allowing for the synergistic use of land for both agriculture and energy production[47]. Aquavoltaics systems, which involve installing photovoltaics on water surfaces, offer advantages such as improved solar panel conversion efficiency due to cooling from water and reduced water evaporation rates of the water surface[48]. Vertical wind turbines are also gaining attention for their potential to minimize land use compared to traditional single-rotor horizontal turbines[49]. To reduce water requirements in photovoltaic and wind turbine systems, treating and reusing wastewater generated during the manufacturing process, implementing efficient water management practices, and selecting materials with minimal water requirements are effective strategies[48].

Countries that have access to seawater and brackish water resources can leverage water desalination technologies to produce high-purity water, which has minimal impact on the overall production cost of hydrogen (<2%)[5]. The limited cost impact is due to the relatively low energy consumption of desalination compared to electrolysis, with desalination requiring only 0.072 $kWh_{el}/kg_{H_2}$[39], while electrolysis consumes 53.8 $kWh_{el}/kg_{H_2}$ (see Suppl. Inf. - S.3 Power production - Table 3.b). The direct capital cost and operating cost of desalination account for approximately 3% and 0.2%, respectively, of an electrolyzer system coupled with reverse osmosis[50]. Desalination, based on reverse osmosis, is already widely implemented in coastal regions with considerable renewable energy potential, as it provides high-quality

2050

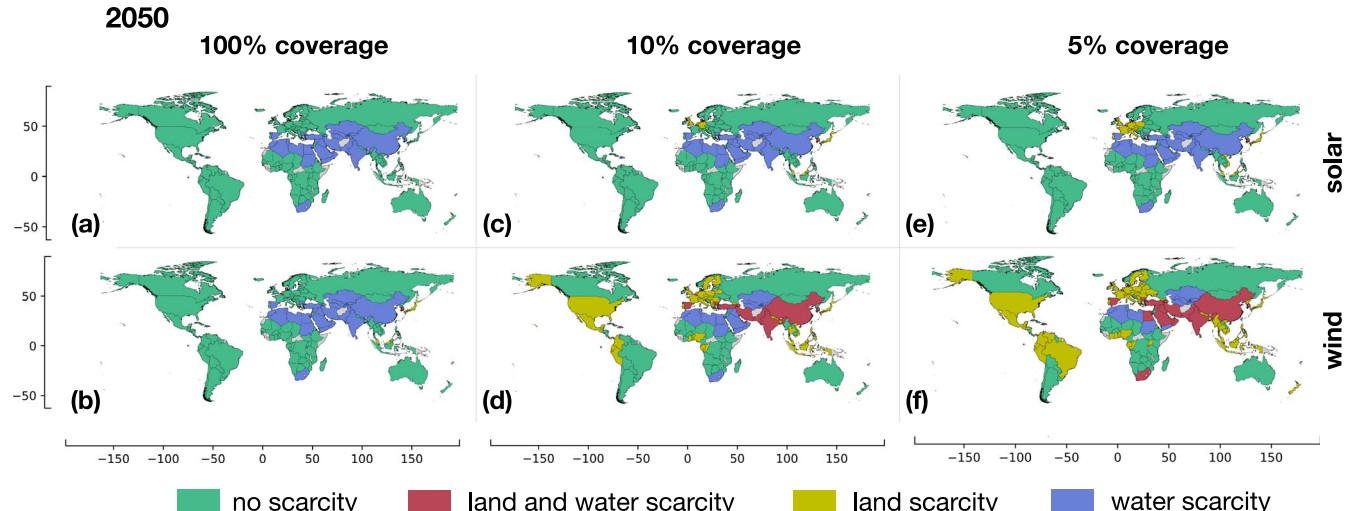

**Fig. 6 | Land and water scarcity induced by hydrogen production in 2050.** Land and water scarcity induced by projected hydrogen production in 2050 for all countries worldwide, considering various fractions of land coverage for **a, c, e** solar and **b, d, f** onshore wind power production. The color-coded regions represent the scarcity status: green for no scarcity, red for both land and water scarcity, yellow for land scarcity only, and blue for water scarcity only. Gray-colored countries indicate unavailable data. The maps are created with the Matplotlib and Geopandas packages for Python[101,102].

deionized water for electrolysis[51]. In land-locked countries, the purification of nontraditional water resources such as municipal and industrial wastewater or brackish water can be considered as an alternative to avoid competition with other water uses[52]. However, it should be noted that the discharge of brine, a highly saline concentrate generated by desalination plants, may have adverse environmental effects due to its high salinity and the presence of treatment chemicals[50]. In this study, solar and wind potential were evaluated separately. Nonetheless, there is an increasing trend towards the adoption of hybrid wind and solar systems, which offer advantages such as maximizing land utilization and reducing storage requirements[53,54] (see results for hybrid system in Suppl. Inf. - Section S.5 Combination of solar and wind). Finally, it should be noticed that all the results presented in this study are consistent with a production of hydrogen only from photovoltaic panels and onshore wind turbines feeding water electrolysis. Other technologies, like nuclear power, hydropower and offshore wind turbines will be part of the technologies for power production in a net-zero scenario. Other technologies like steam methane reforming retrofitted with carbon capture, and pyrolysis, might supply a relevant fraction of the hydrogen demand by 2050 (see "Introduction").

To summarize, although the overall water demand for global hydrogen production (varying between 3.2 and 95.6 billion m³) is minimal compared to the availability of 10,560 billion m³ of water worldwide, it is crucial to identify the countries and regions where hydrogen production may compete with other sectors and exacerbate land and water scarcity. Among these countries, Western European countries, Trinidad and Tobago, South Korea, and Japan stand out as particularly vulnerable in terms of resource scarcity. Trinidad and Tobago, heavily reliant on local oil and gas reserves for its industrial activities, may face challenges in producing the amount of electrolytic hydrogen from renewable sources required to sustain its chemical production, which is a key driver of its economy. Consequently, the country faces the risk of downsizing its chemical industry. Similar risks exist for South Korea and Japan. However, in these two countries a larger portion of hydrogen demand is associated with the transport sector, which could be met through hydrogen imports from other countries. Conversely, regions such as West and Central-East Africa, South America, Canada, and Australia possess abundant resources to meet their hydrogen demand and have the potential to become hydrogen exporters. These regions are favorably positioned to capitalize on their resource availability and seize opportunities in the hydrogen market. While water requirements for hydrogen production is not expected to create water scarcity in countries where it does not already exist (except in Trinidad and Tobago), it does have the potential to exacerbate the unsustainable utilization of water resources. In these regions, it becomes essential to conduct site-specific assessments to evaluate the sustainability of new hydrogen projects in terms of water requirements. While geophysical constraints may not hinder the establishment of an electrolytic hydrogen economy in most countries, the scale of land required to achieve this goal is substantial. However, social, political, and economic factors may render the extensive use of land for hydrogen production unfeasible, particularly in areas with limited land availability, relatively poor wind and solar resources, and/ or expectations of high demand for electrolytic hydrogen.

## Methods

First, based on historical data on energy consumption[55] in 2020, and assuming country-specific energy consumption in 2050 will scale based on the variation of population, we disaggregate hydrogen demand in 2020 and in a reference 2050 net-zero scenario by country and by sector. The sectors we consider include chemicals, cement, refineries and light industry, steel production, and transportation. Country-specific hydrogen demand is derived solely from energy consumption, independently of economic factors such as labor costs, energy costs, or capital investments that may potentially influence industrial relocation between countries (see the sections "Hydrogen demand" and "Electricity demand"). Next, we conduct a detailed spatially-explicit and high-resolution analysis to produce the power production yield of solar and wind energy in each country (see the section "Energy production"). Using the estimated hydrogen demand per country, assuming hydrogen production through electrolysis powered by wind and photovoltaic energy, we quantify the land area required for such production (see the section "Land requirements"). Furthermore, through an ecological footprint analysis[45], we assess the amount of water needed for the manufacturing and operation of solar panels, wind technologies, electrolyzers, as well as the operation of the electrolyzer (see the section "Water requirements"). Finally, we evaluate the impacts of hydrogen production on land and water scarcity in each country worldwide (see the sections "Land scarcity" and "Water scarcity"). To exclude demand variations directly correlated with the impact of COVID-19, country-specific data for 2020 are derived from

country-specific data for 2019 with corrections only based on the variation of population. For detailed data regarding the demand of hydrogen, electricity demand at the country level, and the potential for electricity production from solar panels and wind turbines at both the country and grid levels, please refer to the Supplementary Dataset.

## Hydrogen demand

The country- and sector-specific demand for electrolytic hydrogen in a net-zero economy for 2050 is derived based on a reference scenario of global demand, which amounts to 400 Mt of hydrogen per year[4]. It is assumed that this demand will be met through electrolytic hydrogen production. In 2020, the global demand for hydrogen reached 94 Mt/y[4]. The sectors considered in this scenario include chemicals, cement, refineries and light industry, steel production, and transportation. To determine country- and sector-specific hydrogen demand, we utilize data on final energy consumption and carbon emissions, allowing for the disaggregation of global hydrogen demand (Figure 1 in Suppl. Inf. - Section S.1.1 Hydrogen demand). The electrolytic hydrogen demand considered in this study does not account for the power production sector due to the challenges of accurately predicting the extent to which each country will adopt hydrogen for seasonal storage of electricity from intermittent renewable technologies and as a fuel for power production technologies[56,57]. It is important to note that if electrolytic hydrogen is widely used as a fuel in the power sector, the demand for electrolytic hydrogen could be approximately 25% higher than what is considered in this study[4].

The chemical industry primarily includes high-value chemicals (HVCs) that are used in the production of plastics, synthetic fibers and rubber. However, hydrogen's involvement in the production of HVCs is limited to serving as a feedstock for the production of methanol, which is then used in HVCs manufacturing (12% of methanol used for HVCs in 2017[58]). Consequently, the definition of the chemical industry, in terms of hydrogen usage, pertains specifically to the production of ammonia and methanol[58,59], which includes hydrogen utilization for the production of HVCs as the final product. The demand of hydrogen in the chemical industry is expected to increase along with the rising demands for ammonia and methanol[58,60]. In our reference scenario, the global demand for ammonia in 2020 for chemical production amounts to 189 Mt/y, projected to reach 243 Mt/y by 2050[4]. Approximately 70% of current ammonia production is utilized for fertilizers, with the remaining fraction used for refrigerants, cleaning agents, or the manufacturing of plastics, explosives, and synthetic fibers[5,60]. The demand for fertilizers, historically driven by ammonia, continues to grow mainly in developing economies, while developed economies reach saturation.

However, the production of fuel additives, explosives, and synthetic materials (such as nylon, acrylic fibers, and nitrile rubber) is expected to drive increased ammonia demand[58,60]. In 2020, the global demand for methanol equals 100 Mt/y, projected to rise to 181 Mt/y for chemical production. Methanol finds application in the production of formaldehyde (which accounts for 40% of methanol production in the plastics and textile industries), solvents, fuels, and acetic acid (used in multiple chemical compounds)[61]. The increased demand for methanol is primarily driven by its use as a fuel additive and as an intermediate product for HVCs, projected[58] to grow by 60% by 2050 compared to 2017. Current ammonia and methanol production have similar production routes, which rely on natural gas, respectively the Haber-Bosch process and the methanol synthesis. Low-carbon alternatives involve hydrogen production from water electrolysis or biomass gasification[58,59,62]. Both ammonia and methanol hold potential as fuels or for distributing and storing hydrogen within an ammonia- or methanol-economy[63–65]. Consistent with previous studies[59], we calculate the demand for hydrogen based on the primary feedstocks used in the conversion process, namely natural gas and coal. Country-specific consumption data for natural gas and coal for non-energy use are obtained from historical World Energy Balances published by the

International Energy Agency (IEA)[55]. We quantify the 2020 production of chemicals (ammonia and methanol) per country using the consumption of natural gas and coal in that year, with coefficients of 0.405 ton of chemicals per ton of natural gas ($t_{chemicals}/t_{NG}$) and 0.718 ton of chemicals per ton of coal ($t_{chemicals}/t_{coal}$), respectively[58]. Historical country-specific flows of natural gas and coal are available from the World Energy Balances published by the International Energy Agency (IEA)[55]. We split the chemicals production per country into ammonia and methanol based on regional shares of production[58] per country ($r_c$ for ammonia and $t_c$ for methanol in Figure 1a in Suppl. Inf. - Section S.1.1 Hydrogen demand). According to available literature, ~15% of ammonia production in China is derived from natural gas, while the remaining 85% is from coal. China's coal-based ammonia production accounts for 93% of global ammonia production from coal, followed by South Africa (3%)[60,66,67]. Subsequently, we project ammonia and methanol production in 2050 based on the 2020 values and the projected population variations per country[68] ($g_c$ in Figure 1.1a in Suppl. Inf. - Section S.1.1 Hydrogen demand). The derived country-specific ammonia and methanol values for 2020 and 2050 are normalized to the respective Reference Technology Scenario (RTS) cost-optimal projections for 2020 and 2050[58], with global values reported in Table 1 in Suppl. Inf. - Section S.1.1 Hydrogen demand. Finally, the country-specific demand for hydrogen in 2020 and 2050 is calculated based on the amount of hydrogen required per ton of ammonia (0.18 $t_{hydrogen}/t_{ammonia}$[69]) and per ton of methanol (0.2 $t_{hydrogen}/t_{methanol}$[69]) along with the estimated ammonia and methanol production in the corresponding years (Figure 1a in Suppl. Inf. - Section S.1.1 Hydrogen demand).

The cement industry stands as the largest emitter in the production of non-metallic minerals[70]. Approximately half of the carbon dioxide emissions associated with cement production result from the use of fossil fuels for generating heat[71]. Here, we explore the scenario where fossil fuels are replaced with electrolytic hydrogen. To estimate the country-specific demand for hydrogen in the cement industry, we allocate the global demand for hydrogen in 2020 and 2050 proportionally based on the carbon emissions of each country's cement industry. Historical carbon emissions data at the plant-level are available from the GID Database[72–74]. First, we aggregate the emissions from individual plants in 2020 to the country level ($e_c$ in Figure 1b in Suppl. Inf. - S.1.1 Hydrogen demand). Using this information, we predict the country-specific emissions in the cement industry for 2050 based on projected population variations per country[68] ($g_c$ in Figure 1b in Suppl. Inf. - S.1.1 Hydrogen demand). Then, we compute the country-specific demand of hydrogen in 2020 and 2050 by multiplying the fraction of global emissions in each country ($e_c$ in Figure 1b in Suppl. Inf. - S.1.1 Hydrogen demand) with the corresponding global demand for hydrogen in cement production in 2020 and 2050[4] (Figure 1b in Suppl. Inf. - S.1.1 Hydrogen demand).

Within the petrochemical industry, hydrogen currently finds its primary use in the removal of sulfur through hydrodesulfurization and in the removal of nitrogen compounds through hydrodenitrogenation in hydrocarbons production[75]. Unlike other sectors, the demand for hydrogen in refineries generally decreased in most net-zero-emission energy scenarios due to a reduction in oil consumption[76]. In light industry, hydrogen holds potential for generating medium- and high-temperature heat in the production of various goods such as aluminum, paper, glass, non-metallic minerals, vehicles, machinery, textiles[4,77]. In the steel industry, hydrogen can be utilized both for heat production via combustion in blast furnaces and for iron ore reduction in basic-oxygen and direct reduced iron furnaces[78]. Replacing coke with low-carbon hydrogen for syngas production can result in emissions reductions of 21% in a blast furnace and 60% in a direct reduced iron furnace[78]. Transportation represents the sector with the potential for the largest increase in hydrogen demand, projected to rise from effectively 0 in 2020 to 207 Mt/y in 2050[4,40]. In the realm of road transportation, hydrogen is expected to play a crucial role in heavy-

duty and long-range vehicles, particularly trucks[4]. In maritime transportation, hydrogen can indirectly contribute through methanol[79] or ammonia[80], which possess higher densities. For the aviation sector, sustainable aviation fuels derived from hydrogen or biomass offer a direct replacement for fossil jet fuels in end-use technology and hold significant promise for decarbonizing aviation[81–84]. To estimate hydrogen demand in 2020 and 2050 for these sectors at the country level, we employ a two-step approach. First, we project the total final energy consumption (TFC) per sector in 2050 based on the projected variations per country, using 2020 values as a reference[68] ($g_c$ in Figure 1c in Suppl. Inf. - Section S.1.1 Hydrogen demand). The historical country-specific TFC data for light industry, steel, transport, and refineries are available from the World Energy Balances published by the International Energy Agency (IEA)[55]. Subsequently, we allocate the global hydrogen demand to each country by multiplying the fraction of TFC per country per sector in 2020 and 2050 ($s_c$ in Figure 1c in Suppl. Inf. - Section S.1.1 Hydrogen demand) with the respective global demand of hydrogen per sector in 2020 and 2050 (Figure 1c in Suppl. Inf. - Section S.1.1 Hydrogen demand).

## Electricity demand

The process of obtaining electricity demand for 2020 and 2050 involves two steps. Firstly, we estimate future electricity demand by scaling the country-specific demand in 2019[85] according to the population variations per country[68]. This step allows us to consider the potential differences in economic growth among countries and their corresponding changes in direct electricity consumption resulting from demographic shifts. Secondly, we adjust the estimated demand for each country and year based on the total electricity consumption at the regional level (Europe, Asia-Pacific, Middle-East, Africa, Commonwealth of Independent States, North America, Latin America)[86]. This second step ensures the alignment of electricity consumption with global predictions at the regional level[86]. Additional information and details regarding sample countries can be found in Suppl. Inf. - Section S.2 Electricity demand.

## Energy production

The energy production yield, $S$, is a technology-, time-, space-, and weather-dependent parameter. To quantify this parameter, we adopt a bottom-up approach, utilizing a yearly-average geographical discretization at a grid resolution of 0.75° × 0.75° (equivalent to approximately 80 km × 80 km at the Equator). Specifically, we calculate the energy production per unit area from solar photovoltaics for all grid cells, $N$, using the following formula[87]:

$$S_i^{solar} = \eta^{solar} \gamma I_i \quad \forall\, i \in N \tag{1}$$

with:
- $S_i^{solar}$ (TWh/km²/y) yearly energy production from solar panels per square kilometer in grid cell $i$,
- $\eta^{solar}$ (-) conversion efficiency of solar panels,
- $\gamma$ (-) ground cover ratio, representing the ratio of the surface occupied by the photovoltaic cells to the total area occupied by the solar power plant,
- $I_i$ (TWh/km²/y) yearly average global horizontal irradiation in cell $i$.

We calculate the energy production per unit area from onshore wind turbines for all grid cells, $N$, using the following formula[88]:

$$S_i^{wind} = \eta_i^{wind} c_i P_i \quad \forall\, i \in N \tag{2}$$

with:
- $S_i^{wind}$ (TWh/km²/y) yearly energy production from onshore wind in grid cell $i$,

- $\eta_i^{wind}$ (-) array efficiency of the wind park dependent on the distance between wind turbines[88],
- $c_i$ (-) capacity factor in cell $i$[88],
- $P_i$ (TWh/km²/y) rated power per square kilometer in grid cell $i$.

The energy production (see Figure 2 in Suppl. Inf. - Section S.3 Power production) undergoes a final correction by excluding grid cells that fall within geographical polygons of protected areas[89] or areas with high water stress (indicated by a baseline water depletion indicator >3)[90]. This correction is necessary to prevent bias towards higher power production yields in regions where technology installation would be constrained by the presence of protected areas or water scarcity. It is important to note that this exclusion has a direct impact on the distribution of the power production yield within a country, but is independent from the quantification of water availability and scarcity in a country (see the sections "Water requirements" and "Water scarcity"). For solar production specifically, data limitations exist beyond +60° N and −45° S due to the original dataset on solar irradiation. However, these limitations primarily affect locations with very low solar irradiation[91] (Figure 2 in Suppl. Inf. - S.3 Power production). To combine the grid-cell-specific solar and wind production potentials, various approaches can be employed. In this study, we aggregate energy production at the country level by calculating the area-weighted energy production of cells in the top quartile within each country[92]. This aggregation is representative of the choice to install renewable technologies in locations with high irradiance or rated power. Detailed data on the yearly production of solar panels and wind turbines at the grid-cell level can be found in the Supplementary Dataset and are visualized in Figure 2 in Suppl. Inf. - S.3 Power production.

## Land requirements

The land requirements for the installation of solar panels and onshore wind turbines are determined based on the hydrogen demand calculated in Methods subsection "Hydrogen demand". These land requirements are computed for all the countries, $C$, as follows:

$$A_c = D_c / (S_c \eta^{electrolyzer}) \quad \forall\, c \in C \tag{3}$$

where $D_c$ is the total demand of hydrogen in country $c$; $A_c$ is the land area occupied by solar panels, $A_c^{solar}$, or onshore wind parks, $A_c^{wind}$, required to run water electrolysis in country $c$. $S_c$ is the amount of energy generated per unit of area from solar panels ($S_c^{solar}$) or wind turbines ($S_c^{wind}$); $\eta^{electrolyzer}$ is the conversion efficiency (electricity to hydrogen) of a water electrolyzer[93]. The country-specific values of energy production ($S_c^{solar}$, $S_c^{wind}$) were computed from the grid cells with the top 25% energy production contained within the boundary of country $c$, averaged on the cells area. Values of $\eta^{electrolyzer}$, $S_c^{solar}$, and $S_c^{wind}$ are provided in Suppl. Inf. S.3 for reference countries.

## Water requirements

The water requirements, or water withdrawals, for the manufacturing and operation of solar panels and onshore wind turbines combined with electrolyzers are determined based on the hydrogen demand calculated in the section "Hydrogen demand". These water requirements are computed for all the countries, $C$, as follows:

$$W_c = (w + w^{electrolyzer}) D_c / \eta^{electrolyzer} \quad \forall\, c \in C \tag{4}$$

where $w$ is the amount of water required per unit of electricity generated (t$_{H2O}$/kWh of electricity generated) or consumed (t$_{H2O}$/kWh of electricity consumed), based on an ecological footprint analysis of solar panels[45], $w^{solar}$, wind turbines[45], $w^{wind}$, or electrolyzers[46], $w^{electrolyzer}$. Values of $w^{solar}$, $w^{wind}$, and $w^{electrolyzer}$ are provided in Suppl. Inf. - Section S.4 Water demand.

### Land scarcity

Land scarcity is assessed by considering two types of installation constraints: (i) technical constraints related to specific land coverage which determine the amount of land eligible for use, and (ii) socio- and political-constraints, which define the fraction of eligible land covered with solar or wind technologies.

We consider technical constraints associated with three different land coverages: land covered by forests, land dedicated to agricultural uses, and land dedicated to urban areas[33,34]. For each type of constraint, we consider an eligibility coefficient, $f$, which quantifies the fraction of land that can be technically used to install renewable energy technologies (Table 3.d in Suppl. Inf. - Section S.3 Power production). The fractions of eligible land assumed in this study are based on literature and/or considerations based on typology of technology and constraints:

- forest: a coefficient of 0%[94] is assumed for solar panels, for which the density of trees is considered incompatible with their installation, contrary to wind turbines for which 10%[94] of the surface is assumed eligible for installation within forests in agreement with literature,
- agriculture: the installation of solar panels and wind turbines in agriculture can depend on the typology of land cover. We assumed 10%[87] and 70%[94] in agreement with literature,
- urban: the conservative factor of 25% is assumed as the eligible fraction of artificial surfaces for installation of solar-PV. This value corresponds to the usability factor of residential rooftops[95]. For wind turbines, we assume the conservative factor of 0%, excluding the possibility of installing vertical distributed wind turbines.

The total amount of land eligible for the installation of renewable technologies in a country $c \in C$ is derived from the country's total area, $A_c^{country}$:

$$A_c^{eligible} = A_c^{country} - (1-f^{forest})A_c^{forest} - (1-f^{agriculture})A_c^{agriculture}$$
$$- (1-f^{urban})A_c^{urban} \quad \forall\ c \in C$$
(5)

Depending on the eligibility coefficients, the eligible land is computed for solar panels, $A^{eligible}_{solar,c}$, or wind turbines $A^{eligible}_{wind,c}$. The country's total area, $A_c$ and the areas per country for urban coverage $A_{urban,c}$, forests $A_{forest,c}$, and agriculture $A_{agriculture,c}$, are extracted from reference statistics[33,34].

From the amount of eligible land per country (which accounts for technical constraints on land use) we define the amount of land practically covered by renewable technologies and which accounts for socio- and political-constraints. Accordingly, we introduce the coefficient of eligible land coverage, $f^{coverage}$:

$$A_c^{coverage} = f^{coverage} A_c^{eligible} \quad \forall\ c \in C$$
(6)

The value of $f^{coverage}$ could vary from country to country, and is dependent on socio- and political-constraints, difficult to quantify. In the following, we consider values of $f^{coverage}$ varying between 100% and 1%. The first case represents an ideal scenario in which all the technical solar and wind potential is fully utilized, while the second case represents a more limited installation of solar and wind technologies. By introducing a fraction of land coverage, we were able to identify countries where land scarcity is likely to occur due to a combination of constraints with different characteristics. Utilizing country level statistics[33,34] was chosen over an analysis based on geographical information system, which is commonly used in studies to determine land suitability, but introduces additional uncertainties in representing geographical objects[96].

Land scarcity is computed by comparing the values of land coverage, $A_c^{coverage}$, from Eq. 6, with the land requirements for meeting the hydrogen demand in each country, $A_c$, from Eq. 3:

$$LS_c = A_c/A_c^{coverage} \quad \forall\ c \in C$$
(7)

A scarcity factor greater than one implies that more land for power production is required in a country than the land covered by solar and wind technologies.

### Water scarcity

The water availability in country $c$, $WA_c$, is derived by subtracting environmental flow requirements, $EFR_c$, from the total internal renewable water resources[36], $WR_c$, as:

$$WA_c = WR_c - EFR_c \quad \forall\ c \in C$$
(8)

Here, $EFR_c$ are assumed to be 80% of the total renewable water resources in a country ($WR_c$)[37,38,97]; the remaining 20% represents the fraction of renewable water available for human use without affecting the integrity of downstream water-dependent ecosystems and livelihoods[39]. EFR represents the quantity of water that should be left in the environment to preserve aquatic ecosystems and prevent biodiversity loss[98,99]. The total water demand, $WW_c$, is computed as the sum of the water withdrawals for agricultural, industrial and municipality use[36].

Water scarcity is evaluated by comparing the amount of water withdrawals ($WW_c$) in a country with its water availability[90] ($WA_c$):

$$WS_c = WW_c/WA_c \quad \forall\ c \in C$$
(9)

and the water scarcity induced by electrolytic hydrogen production is computed as:

$$WS_c^+ = (WW_c + WW_c^{hydrogen})/WA_c \quad \forall\ c \in C$$
(10)

A scarcity factor greater than one implies that more water is used in a country than it is supported by the environment and therefore water is unsustainably used, depleting freshwater stocks and environmental flows.

## Data availability

The spatially-explicit and country-specific data generated in this study are provided in the Source data file. The processed data are available at https://doi.org/10.5281/zenodo.8073464[100]. This dataset includes information on the renewable potential, as well as country-specific demand for hydrogen and electricity in both 2020 and 2050.

## Code availability

The complete Python codes used for the calculation and visualization of the results can be accessed in the online repository https://doi.org/10.5281/zenodo.8073464[100]. A description of the content of the repository is provided online.

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

## Acknowledgements

The research of D. Tonelli is funded by SPF Economie - Fonds de transition énergétique. His research stay at the Department of Global Ecology at Carnegie Institution for Science was funded by the Fédération Wallonie Bruxelles (FWB) Mobility Grant. The research of P. Gabrielli is partially funded by the Swiss National Science Foundation (SNSF) Exchange Grant No. 214037, for his research stay at the Department of Global Ecology at Carnegie Institution for Science.

## Author contributions

D.T., L.R., and P.G.: research ideation and design; D.T.: data analysis, data collection, and writing—original draft; P.G. and L.R.: writing—edits and revisions; K.C., A.P., and F.C.: study design and writing supervision.

## Competing interests

The authors declare no competing interests.
