## [Peer Review File · Nature Communications]

Global land and water limits to electrolytic hydrogen production using wind and solar resourcesREVIEWER COMMENTS

Reviewer #1 (Remarks to the Author):

The article presents interesting insights for the future large production of green hydrogen. The data that the authors have produced is important to evaluate the risks and limitation of the expected green hydrogen growth worldwide. These information are particularly useful for policy makers that have to guarantee a sustainable future of natural resources such as freshwater and land . The present study is also of help to identify which countries should be prioritised for the production (and potentially export) of future green hydrogen.

My assessment after reading the article is that it still has large room for improvements that I highlight in the following. Generally speaking, the authors should be more transparent when reporting data, % values etc., being clearer on how they have been calculated or their references.

General consideration: The authors must improve the quality of the language by the support of a professional English proofreader. The language style is sometimes poor and grammar mistakes appears occasionally (e.g., line 67-69). Without a sound language quality, the intended message of the authors can be (and in many cases is) lost. Complex issues requires accurate language to be conveyed!

Introduction.

The general overviewed provided in the introduction is pertinent and exhaustive. Nevertheless, a few points have to be clarified

- I would suggest not to refer to "solar energy" but being more specific and using "Photovoltaic energy", or "solar panels" or simply "PV", as "solar energy" is unspecific and refers to all technologies using sunlight;
- "Net-zero" is an acronym that is generally used to describe a future world where all carbon emitted to the atmosphere is 100% recycled, or where the remaining emissions are compensated by e.g., CCS. The label does not pertain single processes or products, where the label "Carbon-neutral" is generally used. Moreover, it is foreseeable that not all electricity used to produce green hydrogen production will be 100% carbon neutral. Even the latest RED II delegated Act allows for green hydrogen production using 90%RE electricity. For these reasons, I would not recommend the use of "net-zero-emission electricity";
- Line 71-73. It is true that a high methane leakage rate can significantly offset the emission benefits of methane. Nevertheless, assuming that these will be kept low also given to the extended efforts that O&G companies and governments have promised, the major risks linked to blue hydrogen produced via SMR is the rate of CO₂ capture as well as the large energy requirement to operate the CO₂-capture. For a review see also Bauer et al., 2022 – On the climate impact of blue hydrogen production;
- Line 77-78. "...using mature technologies" What is with this intended? I would recommend deleting this as the sentence is already complete;
- Line 85. Remove "removal";
- Line 94-96. The term "unequal" is inappropriate. Please consider using other terms like e.g., "variability";
- Line 99: Substitute "define" with "will define".
- Line 105-106. Please add the "previous literature" at the end of the sentence as references.
- Line 108. It would be appropriate to add, maybe as footnote, what you mean with "spatially-explicit".
- Line 111. I would suggest to add "that is" in front of "independent" to make the sentence clearer.

Results

2.1 Country-Specific hydrogen demand

Line 121. Is here "2050" meant (and not 2020)?

Line 130. Please delete "(Figure 1)"

Line 130-133. The US show an electricity demand for H₂ production shifting from 0.2 until 1.2KwEl/p in 2050, and a "hydrogen demand over electricity demand" from 0.13 to 1.3. This means a six-fold increase in H₂ production against a ten-fold increase of hydrogen over electricity demand. If I understand the overall discourse correctly, I do not see how these data comply with the sentence in line 130-132.

Line 135-138. I do not understand what this sentence means: what are the 14% and the 70% ratios that the authors are referring to?

Figure 1: I am highly confused by relating the statements in the text with what I observe in the figure. This has to do probably with the unclear description of the figure. What does that mean: "hydrogen demand over electricity demand (-)" on the green y-axis on the left hand site? The description in the figure caption "ratio of hydrogen-to-electricity demand" is also unclear to me. I may guess what the authors mean, but I find this not sufficient. I would explain this extensively in the main text. Also, what does "green hydrogen" in the left hand-side axis mean? Green hydrogen has not been defined neither named so far, although, interesting enough, it is included as a key word of the article. Please be consistent with your wording.

Line 138-140. Also here, where I can see the "14 and 17 times of higher electricity demand"? Where do I see in the figure the direct electricity demand for 2020 and 2050 for Trinidad and Tobago that the text is referring to?

2.2. Land requirements

Line 151: please add (in words or with reference to the methodology section) how the eligible land has been calculated.

Line 157. The text refers to Trinidad and Tobago although this country is not reported in the figure. I understand that for scaling reason this country is not added, but this should be written in the main text and not only in the figure caption.

Line 161: Figure 2a only refers to "solar". How can this show data explaining the sentence in lines 160-161 that also refers to wind turbines?

Line 168-169: This sentence has to be rephrased.

2.3 Water requirements

Lines 174-177. Is the "renewable water resources" the "water recharged through the annual hydrological cycle", or "the renewable water resources minus the environmental flow requirements"? Also, in case of the latter, this makes no sense. Please clarify.

Line 179. "See Methods" is not sufficient. Please indicate which paragraph is for this case relevant.

Line 179-180. As far as I understand also from the methodology section, the authors calculate the amount of water needed for the manufacturing of solar panels and wind turbines and add it to the water needed to run the electrolyzers. This implies that the manufacturing of these devices is assumed to take place in the countries where they will be installed, that is generally not the case as, for example, PV are mainly built in China but installed worldwide. To my opinion, this bring a major error for the calculation of the water need for each country. This limitation should be clearly discussed in the article. Also, since LCA studies have been used to calculate the water demand for PV and wind turbines, naming only manufacturing is misleading. Please add also the word "operation". Line 1031-2032: How do you get these numbers?

Line 182. "water scarcity" has not yet been defined. Also, by principle water scarcity only appears by more than 100% use of available renewable water. Therefore, by definition "a water scarcity factor greater than 100%" is conceptually wrong.

Line 181. Add "already" in front of "require", otherwise the meaning of the entire sentence is mainly lost.

Line 192, Why is figure 4 also mentioned here?

Figure 3: The orange and red colours are used to indicate the hydrogen production for the scenarios in 2020 and 2050. Nevertheless, for each country two bars are depicted, each for a different year. This is confusing. Please use either two bars per country, or two different colours to indicate the two different time horizons.

2.4 Land Scarcity

Line 205-216. This section lack of a clear explanation to the coefficients (eligible land, Fcoverage, etc.) before commenting on the different countries. This is due mainly to the poor English. The section should also refer to the methodology section where these coefficients are explained in detail. Also, this section lack of the essential explanation of how land scarcity is defined (0 to 2?), that is only explained in the methodology section.

From the figure itself, the labels "100% eligible land" and so on are misleading. The authors do not intend that 100% of the country area is eligible for wind and PV, but that 100% of the land usable for wind and PV is practically covered with renewables. This should be somehow clearer in the figure. Figure 4. The countries are far too small. Please consider a different layout (e.g., the entire A4 for the 6 figures?) to show the figure appropriately. Also what does the symbol " (-) " in the figure caption means?

2.5 Water scarcity

This section starts, differently from the others, with already commenting on the results of Figure 5 that has not been even introduced. Please be consistent with the style of the other paragraphs. Fig. 6. Also here, please consider to present the maps in bigger dimensions.

2.6 Land and water scarcity

I would suggest to merge this section with the discussion section, as here a combination of previous figures is discussed, and some of the comments here are repeated in the discussion section.

3. Discussion

Line 288-289. This is not pertinent in the discussion section. Please move this to the introduction.

Line 307-325. This section describes some of the limitation of the study. Please consider adding also the fact that a big limitation is that the future water and land availability are based on the current geographical and hydrogeological situation, that might differ greatly in 27 years from now.

Line 327-334. This section is completely disconnected to the focus of the study and to its analysis. Therefore, I would warmly recommend to delete it.

Line 352: Please add on the additional electricity needed to run desalinization of seawater, and on the environmental repercussions of this technique. Naming these with appropriate references would be enough. still, please mention (e.g., in %) how much more energy might be needed.

Line 359-368. The discussion now turns on the carbon emission from manufacturing PV and wind turbines. First of all, this is not described as one of the scope of the article, and no deep analysis has been conducted in this regard. Second, emission estimates here reported are not contextualized, lacking a discussion on their contribution to overall emissions by 2050. Thirds, a conservative scenario o 10% hydrogen leakage is to my opinion absurd, as it would imply an enormous loss of an important resource. For this reason, I suggest to delete this section, together with all reference throughout the text that pertain to the topic of CO2 emissions.

I would also suggest to include in the discussion an overview of technological innovation in the field of PV and wind turbines that could help to reduce the water and land need. Also, it would be interesting to show a figure where the main potential hydrogen exporters are listed, together with their potential hydrogen surplus available for exports.

Since countries like Trinidad and Tobago are mentioned several times in the article, I would suggest to add a longer discussion on why these countries performed particularly well/badly. this doe not need to cover an energy overview of the country, but maybe highlight why when analysing the results they perform they way they do (high denyity population? industrial infrastructure present?).

Conclusions

Line 370-375. This section would more pertain to the introduction rather than conclusions.

Reviewer #2 (Remarks to the Author):

Before publication, the following items should be addressed by the author:

1. English must be improved.
2. The novelty of the paper and the new approach to the topic should be highlighted.
3. The most important quantitative results must be mentioned in the Abstract.
4. The problem statement should be developed.
5. The Journal's standards for referencing must be considered.
6. A list of Abbreviations must be given.
7. To indicate the importance of hydrogen and its role in the future, the following papers are suggested to be considered:
"Hydrogen has found its way to become the fuel of the future." *Future Energy* 1.3 (2022): 11-12.
<https://doi.org/10.55670/fpll.fuen.1.3.2>
"Hydrogen-A sustainable energy carrier." *Progress in Natural Science: Materials International* 27.1 (2017): 34-40.
8. What are the limitations of low-carbon alternatives for hydrogen production, such as coupling of steam methane reforming with carbon capture and renewable natural gas from anaerobic digestion of biomass feedstock?
9. How can large-scale deployment of electrolytic hydrogen affect the availability of land and water resources for food production and other uses?
10. Highlight the main drivers for the increase in hydrogen demand in the chemical sector from 2020 to 2050.
11. Highlight the hydrogen strategies in the countries that will experience the highest increase in per capita demand for hydrogen in the reference case for 2050.
12. Highlight the main regions affected by water scarcity in hydrogen production, and how do they compare to regions with no water scarcity?
13. How does the fraction of eligible land coverage assumed affect the number of countries limited by land and water resources in hydrogen production?
14. Explain the potential consequences for countries limited by both land and water resources in hydrogen production, and what alternatives do they have to achieve net-zero targets?

Reviewer #3 (Remarks to the Author):

This article provides excellent insight on often overlooked considerations in the implementation of a global hydrogen economy, particularly related to land and water scarcity as far back in the process as the production of renewable electricity by solar and wind technologies. While the article agrees with conclusions that are widely accepted in the scientific community that Earth provides an abundance of water needed to meet the stoichiometric needs of water electrolysis, this study considers the very important contributions of solar panel and wind turbine production. The country-by-country breakdown of green hydrogen production capabilities is critical to understanding how and where the hydrogen economy will develop and change the landscape of the global economy. My recommendations to the authors are related to the assumptions and approaches used in these projections, but with the revisions below the article in my opinion is thus appropriate for publication in *Nature Communications*.

Some considerations for the authors during revisions include:

- (1) Some grammar corrections for run-on sentences or other incorrect sentence structure, including:
 - "Constraints varying from the local... transport of methane" (Lines 67-69)
 - "Despite... water resources." (Lines 393-395). Consider using "Although" as an alternative to "Despite"
 - "With the exception of Trinidad... municipal demand) (Lines 227 – 231). The sentence structure here sounds awkward.

(2) Some suggestions for clarity in figures:

- Figure 1: Is the left-hand side y-axis in kW_{el} per person? If so, could you make this unit label agree with Figure 2 (listed as per capita)

(3) In lines 307-310, the authors indicate that the current hydrogen demand is used in the projections for future hydrogen demand in the chemical and petrochemical industries. What factors contribute to the uncertainty and make projections difficult, compared to the projections that are used for other sectors? If so, please reconcile these industries when comparing to other sectors that have undergone projection in the methods (such as ammonia and methanol production projections referenced in lines 460-461).

(4) In line 416, the authors indicate that they corrected data from 2019 to meet the need for 2020 data excluding the impact of COVID-19. Is it feasible to just 2019 data directly as the earlier case study to avoid further approximation?

(5) In section 5.3.1, the authors describe how they determine the energy production yield of solar and wind energy, and in sections 5.3.2 and 5.3.3 they describe the land and water requirements for each of these energy production methods. However, it seems that the cases are divided into countries using 100% solar energy or 100% wind energy for the data in the manuscript. Could the authors consider cases that include a combination of wind and solar resources inside one country?

(6) The authors have decided to omit the power/energy storage sector from their hydrogen demand projections due to uncertainties in the amount of hydrogen allocated for this use in 2050. However, as the authors indicate, this could lead to an increase in the hydrogen demand of 24% by 2050. If there is interest in including this sector, the following sources provide information about projections of the usage of renewables for power in 2050 and the need for about 10-20% of storage as hydrogen.

1. U.S. Energy Information Administration, *International Energy Outlook 2020 (IEO2020)*, 2020.

2. M. Sterner, I. Stadler, *Handbook of Energy Storage Demand, Technologies, Integration*, 2nd ed., Springer, 2019.

3. I. Petkov, P. Gabrielli, Power-to-hydrogen as seasonal energy storage: an uncertainty analysis for optimal design of low-carbon multi-energy systems, *Appl. Energy*. 274 (2020) 115197. <https://doi.org/10.1016/j.apenergy.2020.115197>.

4. A.O. Converse, Seasonal energy storage in a renewable energy system, in: *Proc. IEEE*, 2012: pp. 401-409. <https://doi.org/10.1109/JPROC.2011.2105231>.

Manuscript No.: NCOMMS-23-12575A

We thank the reviewers for their insightful and helpful suggestions.

Below, you will find the point-by-point response incorporating the reviewers' comments (black) along with our corresponding replies (blue). Manuscript text is presented in italics.

We have addressed all reviewer concerns, and believe this review process has improved our manuscript substantially.

REVIEW #1

Remarks to the author

The article presents interesting insights for the future large production of green hydrogen. The data that the authors have produced is important to evaluate the risks and limits of the expected green hydrogen growth worldwide. This information is particularly useful for policy makers that have to guarantee a sustainable future of natural resources such as freshwater and land. The present study is also of help to identify which countries should be prioritized for the production (and potentially export) of future green hydrogen. My assessment after reading the article is that it still has large room for improvements that I highlight in the following. Generally speaking, the authors should be more transparent when reporting data, % values etc., being clearer on how they have been calculated or their references. General consideration: The authors must improve the quality of the language by the support of a professional English proof-reader. The language style is sometimes poor and grammar mistakes appear occasionally (e.g., line 67-69). Without a sound language quality, the intended message of the authors can be (and in many cases is) lost. Complex issues require accurate language to be conveyed!

We have carefully considered the feedbacks and made the necessary revisions to enhance the clarity of the revised version. We believe the updated manuscript meets the level of clarity and addresses the concerns raised by the reviewer.

Introduction

The general overviewed provided in the introduction is pertinent and exhaustive. Nevertheless, a few points have to be clarified.

1. I would suggest not to refer to “solar energy” but being more specific and using “Photovoltaic energy”, or “solar panels” or simply “PV”, as “solar energy” is unspecific and refers to all technologies using sunlight;

We thank the reviewer for the suggestion. We replaced “solar energy” with “solar photovoltaic panels”. (This resulted in changes in many locations in the revised manuscript, e.g., **line 76**).

2. “Net-zero” is an acronym that is generally used to describe a future world where all carbon emitted to the atmosphere is 100% recycled, or where the remaining emissions are compensated by e.g., CCS. The label does not pertain single processes or products, where the label “Carbon-neutral” is generally used. Moreover, it is foreseeable that not all electricity used to produce green hydrogen production will be 100% carbon neutral. Even the latest RED II delegated Act allows for green hydrogen production using 90% RE electricity. For these reasons, I would not recommend the use of “net-zero-emission electricity”;

We thank the reviewer for the suggestion. We addressed this comment with the following modification.

Previous version – **lines 57-58**:

“Electrolytic production of hydrogen using net-zero-emission electricity can contribute^{1,2,3} to achieve net-zero greenhouse gas (GHG) emission goals and keep global warming below 2°C.”

Revised version – **lines 54-55**:

“Electrolytic production of hydrogen using low-carbon electricity can contribute ^{1,2,3} to achieve net-zero greenhouse gas (GHG) emission goals and keep global warming below 2°C.”

3. Line 71-73. It is true that a high methane leakage rate can significantly offset the emission benefits of methane. Nevertheless, assuming that these will be kept low also given to the extended efforts that O&G companies and governments have promised, the major risks linked to blue hydrogen produced via SMR is the rate of CO₂ capture as well as the large energy requirement to operate the CO₂-capture. For a review see also Bauer et al., 2022 – On the climate impact of blue hydrogen production;

We thank the reviewer for raising this point. We addressed this by adding the following sentence at the end of the previous paragraph in the revised version (**lines 62-65**):

“However, these technologies face limitations, such as the local availability of sustainable biomass feedstocks ⁶, infrastructure requirements for carbon capture, transport, and storage^{10,11}, methane leakage management ¹², and limited CO₂ capture rates at the plant level (55%-93%) ⁷.”

Reference number 7 corresponds to Antonini et al., 2020 cited in Bauer et al., 2022.

We cite the review of Bauer et al., 2022, as reference 15, in **lines 68-69** of the revised version when specifically referring to blue hydrogen:

“The leakage rate in the production and transportation system of natural gas is a significant factor affecting the carbon intensity of blue hydrogen ^{14,15,16}.”

4. Line 77-78. “...using mature technologies” What is with this intended? I would recommend deleting this as the sentence is already complete;

We modified the sentence and deleted “using mature technologies”

Previous version, **lines 75-78**:

“While today only 2% of global hydrogen is produced via water electrolysis ⁵, its deployment in combination with an installation of renewable technologies (solar and wind) could allow large-scale production of low-carbon hydrogen using mature technologies.”

Revise version, **lines 71-73**:

“Currently, water electrolysis accounts for only 2% of global hydrogen production ⁵. However, its widespread adoption alongside renewable technologies like solar and wind power could facilitate large-scale production of low-carbon hydrogen.”

5. Line 85. Remove “removal”;

We removed the word “removal” as suggested by the reviewer.

Previous version - **lines 83-85**:

“Wind and solar energy infrastructure can have a significant direct impact on the environment in the areas where it is installed, in terms of habitat removal and biodiversity loss¹⁷.”

Revised version - **lines 89-91**:

“Additionally, the infrastructure required for wind and photovoltaic energy can have significant direct environmental impacts, including habitat loss and biodiversity decline²⁵.”

6. Line 94-96. The term “unequal” is inappropriate. Please consider using other terms like e.g., “variability”;

In response to the reviewer suggestion, we replaced the term “unequal” with “uneven” and “varying”.

Previous version – **lines 94-96**:

“Unequal distribution of water availability combined with unequal economic growth have increased water scarcity and led to a competition in water use between energy, industrial and municipal systems²⁴.”

Revised version – **lines 95-97**:

“The uneven distribution of water availability, coupled with varying economic growth rates among countries, has intensified water scarcity and led to competition in water usage among energy, industrial and municipal systems²⁹.”

7. Line 99: Substitute “define” with “will define”.

This change has been made in accordance with reviewer suggestion. We also modified the sentence by splitting it in two to improve readability.

Previous version – **lines 98-100**:

“However, local and regional water constraints have been recognized among the key factors that define the capability of a country to produce and export electrolytic hydrogen, together with the availability of renewable resources and infrastructure development^{26,27}.”

Revised version – **lines 101-106**:

“However, local and regional water constraints have been recognized among the key factors that define a country’s capability to produce and export electrolytic hydrogen. These constraints are considered alongside the availability of renewable resources and infrastructure development^{19,31}.”

8. Line 105-106. Please add the “previous literature” at the end of the sentence as references.

We added the references as requested by the reviewer, but placed these references on the words “previous literature” to avoid placing a superscripted numerical reference on a numeric value.

Revised version – **lines 111-112**:

“It builds upon previous literature^{4,5} by providing reference cases that disaggregate hydrogen demand by sector and by country for both 2020 and 2050.”

9. Line 108. It would be appropriate to add, maybe as footnote, what you mean with “spatially-explicit”.

We thank the reviewer for the suggestion. A footnote would not be compatible with the formatting requirements from Nature Communications. We added an explanation of what we mean by “spatially-explicit” between parenthesis.

Revised version, **lines 113 – 115**:

“Moreover, this study identifies the potential for electrolytic hydrogen production by combining spatially-explicit data (i.e., high-resolution rasterized geospatial data) on the power production yield of wind and solar technologies with country-specific land availability^{33,34}.”

10. Line 111. I would suggest to add “that is” in front of “independent” to make the sentence clearer.

In the revised version we modified the sentence for better clarity, **lines 116-118**:

“By quantifying the local impact of water demand for hydrogen production at the country level, this research goes beyond assessing the potential impact of hydrogen production on global water resources, which does not account for country-specific constraints¹⁸.”

Results

2.1 Country-specific hydrogen demand

11. Line 121. Is here “2050” meant (and not 2020)?

We thank the reviewer for pointing out to us that our meaning was not clear. We did mean 2020, and that was not a mistake. We have clarified our meaning by changing the text in the following way.

Revised version, **lines 128 – 132**:

“While the hydrogen demand in 2020 is currently met by fossil resources, our 2020 scenario represents a counterfactual situation where all fuel or feedstock inputs to the chemical, cement, refineries and light industry, steel production, and transport sectors are instead provided by electrolytic hydrogen (see Methods - Section 5.1 Hydrogen demand).”

12. Line 130-133. The US show an electricity demand for H₂ production shifting from 0.2 until 1.2 kW_{el}/p in 2050, and a “hydrogen demand over electricity demand” from 0.13 to 1.3. This means a six-fold increase in H₂ production against a ten-fold increase of hydrogen

over electricity demand. If I understand the overall discourse correctly, I do not see how these data comply with the sentence in line 130-132.

We thank the reviewer for the detailed comments.

Based on **Figure 1**, we agree that the US shows an electricity demand for H₂ production varying from 0.2 kW_{el}/p (2020) to 1.2 (2050) kW_{el}/p. This corresponds to a 6-fold increase, in line with the average 7-fold increase of the 35 countries considered. Differently, “hydrogen demand over electricity demand” for the US varies from 0.15 (2020) to 0.7 (2050), corresponding to approx. 5-fold increase. Accordingly, the per capita direct electricity demand can be derived: $0.2/0.15 = 1.33$ kW_{el}/p (2020); $1.2/0.7 = 1.71$ kW_{el}/p (2050). The corresponding variation between 2020 and 2050 is +29%. Although the increase is lower than the average doubling of the direct electricity demand in countries considered, the United States is one of the countries experiencing a greater increase in the per capita electricity demand for hydrogen production than for direct use. I assume that the value of 1.3 kW_{el}/p indicated by the reviewer for the United States comes from referring to the right y-axis of the left panel of the figure, where Trinidad and Tobago and Qatar are plotted. However, this panel uses a different scale, due to the above-the-average values of Trinidad and Tobago.

In response to feedback from multiple reviewers, we have implemented substantial modifications in the text describing the figure, the y-axis labels of the figure, as well as its caption. These revisions were made with the intention of improving the clarity of the description of **Figure 1**. We hope that these modifications successfully enhanced the understanding of **Figure 1** and provided a clearer representation of the data.

Previous version of main text, **lines 126-140**:

“Figure 1 presents the breakdown of electricity demand needed to produce electrolytic hydrogen in each sector. The difference between 2020 and 2050 is substantial, and is driven by a 80% reduction of hydrogen demand in refineries, a three times increase in hydrogen demand in the chemical sector (ammonia and methanol production) and an increase in the transport sector from zero to more than 200 Mt/y globally (Figure 1). From 2020 to 2050, all countries in Figure 1 are projected to experience a greater increase in the per capita demand of hydrogen in our reference case (on average 7 times higher demand in 2050 than in 2020) compared to the variation of the per capita direct electricity demand (on average 2 times higher in 2050 than in 2020). This is particularly evident for the United States, Brazil and Mexico, while exceptions include Malaysia, South Korea and the Netherlands, where the rise in hydrogen demand is comparable or lower than the rise in electricity demand. The United States in particular would have required a capacity of 0.2 kW per person for hydrogen production in 2020 if the hydrogen were made electrolytically, which increases to 1.2 kW per person in 2050, corresponding to 14% and 70% of the capacity required for direct electricity demand. The intense industrial activity of Trinidad and Tobago, largely driven by their fossil fuel industry, makes it an outlier, with the electricity demand for hydrogen production in our reference cases being 14 to 17 times higher than the direct electricity demand in 2020 and 2050, respectively.”

Revised version of main text, **lines 134-153**:

“Figure 1 illustrates the electricity demand necessary for electrolytic hydrogen production across different sectors. The left y-axis represents the per capita capacity required for electrolytic hydrogen production, while the right y-axis compares the per capita capacity for electrolytic hydrogen with the per capita capacity for direct electricity consumption. There is a significant difference between 2020 and 2050, primarily driven by an 80% reduction in hydrogen demand in refineries, a threefold increase in hydrogen demand in the chemical sector (ammonia and methanol production), and the transport sector transitioning from zero to over 200 Mt/y globally.

In our reference scenario, all countries depicted in Figure 1 experience a larger increase in per capita hydrogen demand from 2020 to 2050 (on average seven times higher demand in 2050 than in 2020) compared to the variation in the per capita direct electricity demand (on average two times higher in 2050 than in 2020). This trend is particularly noticeable in the United States, Brazil, and Mexico. However, exceptions include Malaysia, South Korea, and the Netherlands, where the rise in hydrogen demand is similar or lower than the increase in electricity demand. Specifically, in 2020, the United States would have required a capacity of 0.2 kW per person for hydrogen production if electrolytic hydrogen was used. This capacity would subsequently increase to 1.2 kW per person by 2050. This corresponds to 15% and 70% of the capacity needed for direct electricity demand, respectively. Trinidad and Tobago, known for its extensive fossil fuel industry, stands out as an outlier with required capacity for electrolytic hydrogen production in our reference cases being 14 to 17 times higher than the capacity for direct electricity demand in 2020 and 2050, respectively.”

Previous version of caption **Figure 1, lines 143-147:**

“Figure 1: Hydrogen demand per sector, for selected countries (top 35 countries by demand) in 2020 (a) and 2050 (b). Electricity for hydrogen demand per sector (stacked bars, left-hand side y-axis) and ratio of hydrogen-to-electricity demand (green markers, right-hand side y-axis) in 2020 and in 2050. Conversion is based on $1 \text{ kW}_{el} = 0.62 \text{ kW}_{H_2}$, $1 \text{ kWh}_{H_2} = 0.03 \text{ kg}_{H_2}$, $1 \text{ kW}_{el} = 8760 \text{ kWh}_{el}/y$. The order of the countries reflects the highest total demand of hydrogen in 2050 (except for Trinidad and Tobago and Qatar).”

Revised version of caption **Figure 1, lines 156-166:**

“Figure 1: Hydrogen demand and capacity comparison for selected countries in (a) 2020 and (b) 2050. The figure illustrates the hydrogen demand per sector for the top 35 countries ranked by country-specific demand in 2050. The stacked bars on the left-hand side y-axis represent the per capita capacity required for electrolytic hydrogen production in each sector. The green markers on the right-hand side y-axis represent the ratio of the capacity required for electrolytic hydrogen production to the capacity required for direct electricity demand. To derive the capacity for direct electricity demand (kW_{el} per capita), divide the capacity for electrolytic hydrogen (y-axis, left-hand side) by the corresponding ratio of the capacity for electrolytic hydrogen to direct electricity demand (y-axis, right-hand side). The conversion factors used are: $1 \text{ kW}_{el} = 0.62 \text{ kW}_{H_2}$, $1 \text{ kWh}_{H_2} = 0.03 \text{ kg}_{H_2}$, and $1 \text{ kW}_{el} = 8760 \text{ kWh}_{el}/y$. The countries are ordered based on their total hydrogen demand in 2050, apart from Trinidad and Tobago and Qatar. Note: the Supplementary Dataset provides country-specific results for hydrogen demand and capacity.”

Previous version of **Figure 1:**

Revised version of Figure 1:

13. Line 135-138. I do not understand what this sentence means: what are the 14% and the 70% ratios that the authors are referring to?

We thank the reviewer for pointing out our need for greater clarity.

In agreement with the previous comment, the US shows an electricity demand for H₂ production varying from 0.2 kW_{el}/p (2020) to 1.2 (2050) kW_{el}/p. The per capita direct electricity demand is: 0.2/0.15 = 1.33 kW_{el}/p (2020); 1.2/0.7 = 1.71 kW_{el}/p (2050). The comparison of the two capacities in the two years gives: 0.2/1.33*100 = 15% (2020); 1.2/1.71*100 = 70% (2050). We corrected the percentage for the United States.

Previous version, **lines 135-138**:

“The United States in particular would have required a capacity of 0.2 kW per person for hydrogen production in 2020 if the hydrogen were made electrolytically, which increases to 1.2 kW per person in 2050, corresponding to 14% and 70% of the capacity required for direct electricity demand.”

Revised version, **lines 147-150**:

“Specifically, in 2020, the United States would have required a capacity of 0.2 kW per person for hydrogen production if electrolytic hydrogen was used. This capacity would subsequently increase to 1.2 kW per person by 2050. This corresponds to 15% and 70% of the capacity needed for direct electricity demand, respectively.

14. Figure 1: I am highly confused by relating the statements in the text with what I observe in the figure. This has to do probably with the unclear description of the figure. What does that mean: “hydrogen demand over electricity demand (-)” on the green y-axis on the left-hand side? The description in the figure caption “ratio of hydrogen-to-electricity demand” is also unclear to me. I may guess what the authors mean, but I find this not sufficient. I would explain this extensively in the main text. Also, what does “green hydrogen” in the left hand-side axis mean? Green hydrogen has not been defined neither named so far, although, interesting enough, it is included as a key word of the article. Please be consistent with your wording.

We thank the reviewer for the detailed comments.

Different updates of the text and the figures led to the inconsistencies highlighted. We replaced “hydrogen demand over electricity demand (-)” in the figure with “(capacity hydrogen)/(capacity direct electricity) (-)”. We also replaced the description “ratio of hydrogen-to-electricity demand” in the caption of the figure with “ratio of the capacity required for electrolytic hydrogen production over the capacity required for direct electricity demand”. The following sentence was added in the main text for better clarification, **lines 135 - 137** of the revised version:

“The left y-axis represents the per capita capacity required for electrolytic hydrogen production, while the right y-axis compares the per capita capacity for electrolytic hydrogen with the per capita capacity for direct electricity consumption.”

We replaced the term “green hydrogen” with “electrolytic hydrogen” in the figure and among the keywords. We modified the previous label of the y-axis on the left-hand side, “electricity for green hydrogen (kW_{el}/p)”, with “electrolytic hydrogen capacity (kW_{el} per capita)”.

15. Line 138-140. Also here, where I can see the “14 and 17 times of higher electricity demand”? Where do I see in the figure the direct electricity demand for 2020 and 2050 for Trinidad and Tobago that the text is referring to?

We understand that the inconsistency highlighted in the previous comments led to confusion in the interpretation of the figure. We modified the sentence in the following form (revised version, **lines 150-153**):

“Trinidad and Tobago, known for its extensive fossil fuel industry, stands out as an outlier with required capacity for electrolytic hydrogen production in our reference cases being 14 to 17 times higher than the capacity for direct electricity demand in 2020 and 2050, respectively.”

These values can be seen from the y-axis, left-hand side, in the left panel related to Trinidad and Tobago and Qatar. Due to the log scale of the plot, the last two ticks of the scale correspond to 10 and 20. The specific value can also be found in the dataset of the supplementary information.

2.2 Land requirements

16. Line 151: please add (in words or with reference to the methodology section) how the eligible land has been calculated.

The definition of eligible land is presented in section **Methods – 5.4.1 Land scarcity**. We highlighted the definition in **lines 168 – 174** of the revised manuscript:

“Figure 2 (a, b) presents the per capita land requirements to meet hydrogen demand in our reference cases and compares it with the eligible land (as defined in Methods - Section 5.4.1 Land scarcity) in selected countries for 2020 and 2050. The eligible land of each country is determined by subtracting the fractions of the area occupied by agriculture, artificial surfaces, and forests (which are considered ineligible for the installation of renewable energy systems) from the total area of the country. This calculation helps identify the portion of land that can be utilized for the installation of renewable energy infrastructure, such as solar panels and wind turbines, to meet energy demands sustainably.”

17. Line 157. The text refers to Trinidad and Tobago although this country is not reported in the figure. I understand that for scaling reason this country is not added, but this should be written in the main text and not only in the figure caption.

We added the comment between parenthesis in the revised version, **lines 185-187**:

“Trinidad and Tobago (not shown in Figure 2 for scaling reasons) has sufficient available land for hydrogen production only when considering electricity production from solar panels and the hydrogen demand of 2020.”

18. Line 161: Figure 2a only refers to “solar”. How can this show data explaining the sentence in lines 160-161 that also refers to wind turbines?

We thank the reviewer for spotting this mistake. We wanted to refer only to “solar”. We removed “and wind turbines” from the sentence in **lines 187-189**:

“Conversely, several countries, including Canada, Australia, Russia, Algeria and Argentina, would only need to cover 1% of their eligible land with solar panels to meet their projected hydrogen demand in 2050 (Figure 2 (a)).”

19. Line 168-169: This sentence has to be rephrased.

We thank the reviewer for once again pointing where our expression could be improved. We rephrased and extended the sentence in **lines 206-208**:

“The gray areas (and pink areas) represent fractions (and multiples) of the eligible land, as defined in Methods - Section 5.4.1 Land scarcity, and are used for comparison with the land requirements for hydrogen demand in each country.”

2.3 Water requirements

20. Lines 174-177. Is the “renewable water resources” the “water recharged through the annual hydrological cycle”, or “the renewable water resources minus the environmental flow requirements”? Also, in case of the latter, this makes no sense. Please clarify. Line 179. “See Methods” is not sufficient. Please indicate which paragraph is for this case relevant.

We thank the reviewer for the comments related to **lines 174-177** and **179**. We addressed this concern through changes on what is now **lines 224-227**.

The figure compares the two metrics which appear in the definition of water scarcity per country, according to **Equation 9** in **Section 5.4.2 Water scarcity**, part of **Methods**:

$$WS_c = WW_c/WA_c$$

where:

- WW_c denotes water withdrawals in agriculture, industry, municipality, and hydrogen production,
- WA_c denotes renewable water available, which is quantified as the water from the annual hydrological cycle of a country minus the water flows required to sustain water-dependent ecosystems and livelihoods at country level (**Eq. 8** in **Section 5.4.2 Water scarcity**)
-

The figure aims at providing a breakdown of the water withdrawals per sector and comparing the water use for hydrogen production with water use in agriculture, industry and municipality. We modified the text to provide more clarity in the metrics presented in **Figure 3**. We specified that water availability was defined in **Equation 8** of **Section 5.4.2 Water scarcity**. We highlighted how the water withdrawals and water availability define water scarcity from **Equation 10** of **Section 5.4.2 Water scarcity**.

Revised version, **lines 219-223**:

“Figure 3 (a, b) also compares these withdrawals with available renewable water resources in selected countries, as defined in Methods (Section 5.4.2 - Water scarcity - Equation 8), for the years 2020 and 2050. The height of the bars represents the numerator in the calculation of water scarcity per country (see Methods - 5.4.2 Water scarcity - Equation 10), while the contours in the background represent the denominator.”

21. Line 179-180. As far as I understand also from the methodology section, the authors calculate the amount of water needed for the manufacturing of solar panels and wind turbines and add it to the water needed to run the electrolyzers. This implies that the manufacturing of these devices is assumed to take place in the countries where they will be

installed, that is generally not the case as, for example, PV are mainly built in China but installed worldwide. To my opinion, this brings a major error for the calculation of the water need for each country. This limitation should be clearly discussed in the article. Also, since LCA studies have been used to calculate the water demand for PV and wind turbines, naming only manufacturing is misleading. Please add also the word “operation”. Line 1031-2032: How do you get these numbers?

We thank the reviewer for the comment. We now include this point in the **Discussion, lines 410 – 416**:

“An additional assumption of this study is the production of solar panels and wind turbines to occur within the same country as their installation sites. While this assumption simplifies the analysis, it does not fully capture the complexities of global supply chains behind these technologies. However, one of the main findings of this work is the relatively minor impact of water demand for low-carbon hydrogen production in the creation of water scarcity (with the exception of Trinidad and Tobago). Accordingly, we do not expect that incorporating a model of the supply chain for photovoltaic panels and wind turbine supply would materially alter the conclusions drawn from our work.”

We added the word “operation” in the revised version, **lines 228, 427, 501, 628, 687**.

The numbers in **lines 1031-1032** of the previous version were derived from the following values:

year hydrogen demand scenario	electricity demand hydrogen production (TWh _{el} /y)	water demand photovoltaics (10 ⁶ m ³ /TWh _{el})	water demand wind turbines (10 ⁶ m ³ /TWh _{el})	water demand hydrogen photovoltaics (10 ⁹ m ³ /y)	water demand hydrogen wind turbines (10 ⁹ m ³ /y)
2020	4695	2.75	0.65	14	3
2050	21520	2.75	0.65	59	14

- the electricity demand for hydrogen production is computed from the hydrogen demand in **Methods – 5.1 Hydrogen demand**, corresponding to 94 Mt/y in 2020 and 400 Mt/y in 2050. An electrolyzer efficiency of 0.62 is assumed accordingly to **Table S.3.c in Supplementary Information – S.3 Power production**
- coefficients of water consumption per technology (photovoltaics and wind turbines) are reported in **Table S.4 in Supplementary Information – S.4 Water use**

Since we noticed that the assumption of hydrogen demand for 2020 was not explicitly mentioned in **Methods – 5.1 Hydrogen demand**. We added the following sentence in **lines 513-514**: “In 2020, the global demand for hydrogen reached 94 Mt/y⁴.”.

22. Line 182. “water scarcity” has not yet been defined. Also, by principle water scarcity only appears by more than 100% use of available renewable water. Therefore, by definition “a water scarcity factor greater than 100%” is conceptually wrong.

We agree with the reviewer. Accordingly, we removed the sentence “(i.e., a water scarcity factor greater than 100%)” which could be misleading, instead of providing clarifications.

23. Line 181. Add “already” in front of “require”, otherwise the meaning of the entire sentence is mainly lost.

The sentence in the revised version reads (lines 229-231):

“Independently of the electricity generation technology, several countries, which include Saudi Arabia, Algeria, Trinidad and Tobago, China, India, Egypt, Turkey, and South Korea, already exceed their sustainable domestic water resources.”

24. Line 192, Why is figure 4 also mentioned here?

This is a mistake. We wanted to refer to **Figure 3** and **Figure 5**, since **Figure 5** focuses on the exacerbation of water scarcity. To avoid confusion in anticipating the content of **Figure 5**, we eliminated the reference to the figure.

25. Figure 3: The orange and red colours are used to indicate the hydrogen production for the scenarios in 2020 and 2050. Nevertheless, for each country two bars are depicted, each for a different year. This is confusing. Please use either two bars per country, or two different colours to indicate the two different time horizons.

We thank the reviewer for this remark. We preferred to keep two different bars per country, one per year, and use a single color. We added a magnification of two columns to highlight that they refer to two different years. We did the same modification for **Figure 2**.

Previous version of **Figure 2**:

Revised version of Figure 2:

Previous version of Figure 3:

Revised version of Figure 3:

2.4 Land scarcity

26. Line 205-216. This section lack of a clear explanation to the coefficients (eligible land, F coverage, etc.) before commenting on the different countries. This is due mainly to the poor English. The section should also refer to the methodology section where these coefficients are explained in detail. Also, this section lack of the essential explanation of how land scarcity is defined (0 to 2?), that is only explained in the methodology section.

We thank the reviewer for the comment and apologize for the lack of clarity. We extended the first part of the section related to the description of **Figure 4** and we explicitly indicated the reference to the section of the methods.

Previous version, **lines 205-207**:

“Figure 4 shows the degree of land scarcity in all countries for different fractions of eligible land coverage ($f^{coverage}$) considered for the installation of solar and onshore wind technologies in our reference cases.”

Revised version, **lines 258-266**:

“Figure 4 depicts the extent of land scarcity across all countries, considering varying fractions of eligible land coverage (f coverage) for the installation of solar and onshore wind technologies in our reference cases, as defined in Methods - Section 5.4.1 Land scarcity. The eligible land of a country refers to the total land area that can be technically covered with renewable technologies (f coverage = 100%). However, additional economic and socio-political constraints limit the practical coverage of land with renewable technologies, which is quantified by the fraction of eligible land coverage (f coverage < 100%). By comparing the land required for electrolytic hydrogen production with the land practically covered by renewable technologies (LS in Methods - Section 5.4.1 Land scarcity - Equation 7), we assess the presence (LS > 1) or the absence (LS < 1) of land scarcity resulting from hydrogen production.”

27. From the figure itself, the labels “100% eligible land” and so on are misleading. The authors do not intend that 100% of the country area is eligible for wind and PV, but that 100% of the land usable for wind and PV is practically covered with renewables. This should be somehow clearer in the figure.

We thank the reviewer for underlining the need for this correction.

We made the labels of **Figure 4** consistent with those of **Figure 2**. In modifying the figure, we also noticed a small discrepancy in the color scale and we updated the plots. We apologize for the mistake in the previous version of the figure.

Previous version of **Figure 4**:

New version of **Figure 4**:

28. Figure 4. The countries are far too small. Please consider a different layout (e.g., the entire A4 for the 6 figures?) to show the figure appropriately. Also what does the symbol “(-)” in the figure caption mean?

We thank the reviewer for the suggestion.

We used the A4 layout as suggested. The symbol “(-)” was referring to the unit of measure of the “land scarcity” metric, equal to the ratio of two areas. However, we recognize that it does not add any relevant information to the figure, so we removed it. Following the previous comment on **Figure 4**, we also made the labels of **Figure 6** consistent with those of **Figure 2**. In modifying the figure we also noticed an error in plot (d), that we updated.

Previous version of **Figure 6**:

New version of **Figure 6**:

2.5 Water scarcity

29. This section starts, differently from the others, with already commenting on the results of Figure 5 that has not been even introduced. Please be consistent with the style of the other paragraphs.

We thank the reviewer for the remark. We modified the section in order to be consistent with the style of the previous paragraphs.

Previous version, **lines 227 – 233**:

“With the exception of Trinidad and Tobago in case of hydrogen production from solar panels, in our reference cases, the demand of water for hydrogen production does not create water scarcity anywhere in the world when water scarcity is not already present, being water withdrawals for hydrogen production negligible compared to the sum of the water demand in other sectors (agriculture, industry and municipal demand) (Suppl. Inf. S.4). However, in 37 countries with existing water scarcity, hydrogen production leads to scarcity exacerbation. Figure 5 highlights these countries by showing the increase in water scarcity compared to the case with no hydrogen production (see Methods - Equations 9, 10).”

Revised version, **lines 286 – 291**:

“Figure 5 highlights the exacerbation in water scarcity compared to the scenario without hydrogen production (see Methods - Section 5.4.2 - Equations 9, 10). Except for Trinidad and Tobago, where hydrogen production from solar panels can lead to water scarcity, the assumed reference scenario for hydrogen demand does not create water scarcity anywhere

in the world if water scarcity is not already present (the relative increase in water withdrawals due to hydrogen production is shown in Suppl. Inf. – Section S.4.2 Impact of water requirements for hydrogen production).”

30. Fig. 6. Also here, please consider to present the maps in bigger dimensions.

We increased the size of the figure. Since the figure contains only two maps, we prefer to adopt this solution compared to expanding the size to A4.

2.5 Land and water scarcity

31. I would suggest to merge this section with the discussion section, as here a combination of previous figures is discussed, and some of the comments here are repeated in the discussion section.

We thank the reviewer for this comment. We converted **Section 2.6 Land and water scarcity** into **Section 3 Discussion** and we removed the repetition by deleting **lines 288 - 292** of the previous version.

3. Discussion

32. Line 288-289. This is not pertinent in the discussion section. Please move this to the introduction.

We thank the reviewer for sharing this opinion. We find important to underly that a reference scenario of 400 Mt/y represents a key assumption in the derivation of our results with respect to the total requirements of land and water. However, it is worth noting that other scenarios exist and could have been equally considered in the derivation of our results. Accordingly, we prefer to keep this information in the **Discussion** section to avoid the risk of limiting the information to the context of the study.

Revised manuscript, **lines 311 – 313**:

“The predictions for hydrogen demand in 2050 vary significantly across scenarios, ranging from 3% to 12% of the global total final energy consumption, corresponding to a range of 83 Mt/y to 666 Mt/y 40. In on our reference scenario with hydrogen demand of 400 Mt/y in 2050, we find that [...].”

33. Line 307-325. This section describes some of the limitation of the study. Please consider adding also the fact that a big limitation is that the future water and land availability are based on the current geographical and hydrogeological situation, that might differ greatly in 27 years from now.

Thank you, this is a very important remark that we are glad to add to the discussion. We added the following sentences in **lines 406-409**:

“The assumptions made in this study consider current values of water and land availability, which are influenced by existing geographical and hydrogeological conditions. It is important to acknowledge that these conditions have the potential to undergo significant changes over the 27-year period from the present time.”

34. Line 327-334. This section is completely disconnected to the focus of the study and to its analysis. Therefore, I would warmly recommend to delete it.

We thank the reviewer for the comment. We deleted the section, as suggested.

35. Line 352: Please add on the additional electricity needed to run desalination of seawater, and on the environmental repercussions of this technique. Naming these with appropriate references would be enough. still, please mention (e.g., in %) how much more energy might be needed.

We thank the reviewer for the remark. We added the suggested information in the same paragraph of the discussion.

Revised version, **lines 438 – 442:**

“The limited cost impact is due to the relatively low energy consumption of desalination compared to electrolysis, with desalination requiring only 0.072 kWh_{el}/kg_{H2}³⁹, while electrolysis consumes 53.8 kWh_{el}/kg_{H2} (see Suppl. Inf. - S.3 Power production - Table S.3.b). The direct capital cost and operating cost of desalination account for approximately 3% and 0.2%, respectively, of an electrolyzer system coupled with reverse osmosis⁵⁰.”

Revised version, **lines 456 – 458:**

“However, it should be noted that the discharge of brine, a highly saline concentrate generated by desalination plants, may have adverse environmental effects due to its high salinity and the presence of treatment chemicals⁵⁰.”

36. Line 359-368. The discussion now turns on the carbon emission from manufacturing PV and wind turbines. First of all, this is not described as one of the scope of the article, and no deep analysis has been conducted in this regard. Second, emission estimates here reported are not contextualized, lacking a discussion on their contribution to overall emissions by 2050. Thirds, a conservative scenario o 10% hydrogen leakage is to my opinion absurd, as it would imply an enormous loss of an important resource. For this reason, I suggest to delete this section, together with all reference throughout the text that pertain to the topic of CO2 emissions.

We welcomed the comment of the reviewer and deleted the section as suggested.

37. I would also suggest to include in the discussion an overview of technological innovation in the field of PV and wind turbines that could help to reduce the water and land need.

We thank the reviewer for the suggestion. We extended the **Discussion** section with the following text.

Revised version, **lines 434 – 444:**

“Regarding land demand in photovoltaic and wind turbine systems, it can be mitigated through design optimization, the use of innovative materials, and careful site selection. Furthermore, solar panels can be installed in agrivoltaic systems, allowing for the synergistic use of land for both agriculture and energy production⁴⁷. Aquavoltaics systems, which involve installing photovoltaics on water surfaces, offer advantages such as improved solar panel conversion efficiency due to cooling from water and reduced water

evaporation rates of the water surface⁴⁸. Vertical wind turbines are also gaining attention for their potential to minimize land use compared to traditional single-rotor horizontal turbines⁴⁹. To reduce water requirements in photovoltaic and wind turbine systems, treating and reusing wastewater generated during the manufacturing process, implementing efficient water management practices, and selecting materials with minimal water requirements are effective strategies⁴⁸.”

38. Also, it would be interesting to show a figure where the main potential hydrogen exporters are listed, together with their potential hydrogen surplus available for exports.

We thank the reviewer for the suggestion. We added a figure with the countries having the larger surplus and deficit in **Supplementary Information - S.9 Sensitivity number of countries with land and water scarcity**.

39. Since countries like Trinidad and Tobago are mentioned several times in the article, I would suggest to add a longer discussion on why these countries performed particularly well/badly. This does not need to cover an energy overview of the country, but maybe highlight why when analysing the results, they perform the way they do (high density population? industrial infrastructure present?).

We expanded the section **Discussion**.

Revised version, **lines 390 – 406**:

“In our findings, Trinidad and Tobago appears as an outlier in terms of the impact of hydrogen demand on land and water requirements. Despite having an estimated population of only around 1 million people, Trinidad and Tobago is one of the world's largest exporters of ammonia and methanol. The country's ability to maintain its leading position as an exporter of chemical products in a net-zero economy hinges on its capacity to achieve a production level of 165 TWh/year of low-carbon hydrogen by 2050, as per our reference scenario. Our results underscore that meeting this required production capacity of low-carbon hydrogen necessitates a combined and highly intensified approach involving the installation of solar panels, onshore, and offshore wind power technologies for electrolytic hydrogen production. The country can also leverage its expertise from the oil and gas industry and its advantageous access to depleted oil and gas fields to continue harnessing its gas deposits for low-carbon hydrogen production, utilizing carbon capture and storage technologies^{6,7}. In other countries with high per capita hydrogen demand and limited renewable resources, such as Japan, South Korea, and various European countries, land scarcity arises from the limitation in eligible land for the installation of solar panels and wind turbines. Among these countries, Japan is planning to import 0.3 million tons per year (Mt/y) of hydrogen, establishing an international supply chain through collaborations with Australia and Brunei⁴². Additionally, the European Union has set a target of importing 50% of its hydrogen supply by 2030⁴³, with Belgium specifically aiming to import 100% of its hydrogen supply⁴⁴ (see Suppl. Inf. - S.1.3 Country-specific hydrogen strategies).”

4. Conclusions

40. Line 370-375. This section would more pertain to the introduction rather than conclusions.

We thank the reviewer for the comment. Those parts of the content that were not redundant were integrated into the introduction, as suggested.

REVIEW #2

1. English must be improved.

We have implemented significant revisions and went through professional proofreading to enhance the writing style of the text.

2. The novelty of the paper and the new approach to the topic should be highlighted.

We thank the reviewer for this remark. We hope that the updated text in the revised version, **lines 77-82**, better highlights the novelty of our work:

“Existing analyses^{18,19,20,21} overlook or consider only global-scale assessments of land and water availability, independent of country-specific limits, when examining the production and trade of hydrogen from countries with abundant renewable resources to those with limited renewable energy potential. In this study, we focus on assessing the global demand and availability of land and water resources at the country level for prospective large-scale electrolytic hydrogen production using wind and solar electricity.”

3. The most important quantitative results must be mentioned in the Abstract.

We thank the reviewer for the remark. We mentioned the most important quantitative results can now be found in the **Abstract**.

Previous version, **Abstract**:

“Scale up of electrolytic production of hydrogen has been proposed as key to achieving net-zero emissions by 2050. This poses technical, economic, and environmental challenges. One challenge concerns the use of wind and solar resources to power water electrolyzers for low-carbon hydrogen production, which results in additional demand for already scarce land and water resources. Here, we quantify the land and water scarcity induced by large-scale hydrogen production from electrolysis of water using wind and solar electricity. First, we develop a reference scenario for hydrogen demand in a 2050 net-zero economy by country and by sector. We then estimate land and water demands associated with future hydrogen production, and compare such demands with the respective country-specific land and water availability. Results highlight that land and water availability may constrain the production of electrolytic hydrogen in Europe, Asia, and Northern Africa. In our reference scenario, less than half of 2050 hydrogen demand could be satisfied by domestic production with no constraints in land and water resources. Our findings identify potential importers and exporters of hydrogen based on domestic land and water resources.”

Revised version, **Abstract**:

“Scale-up of electrolytic hydrogen production has been proposed as key to achieve net-zero emissions by 2050, but it poses technical, economic, and environmental challenges. One such challenge is for policymakers to ensure a sustainable future for natural resources like freshwater and land while facilitating low-carbon hydrogen production using wind and solar power. We establish a country-by-country reference scenario for hydrogen demand in 2050 and compare it with land and water availability. Our analysis highlights the countries that will be constrained by domestic natural resources to achieve electrolytic

hydrogen self-sufficiency in a net-zero target. Depending on land allocation for the installation of solar panels or wind turbines, less than 50% of hydrogen demand in 2050 could be met through a local production without land or water scarcity. Our findings identify potential importers and exporters of hydrogen or, conversely, exporters or importers of industries that would rely on electrolytic hydrogen. The abundance of land and water resources in Southern and Central-East Africa, West Africa, South America, Canada, and Australia makes them potential leaders in hydrogen exports.”

4. The problem statement should be developed.

We thank the reviewer for the remark. We have improved our problem statement. The sentences of the paragraph below summarize well the lack of knowledge that we identified (problem), the motivation for covering this gap with our work (need), and ultimately the significance of our work.

Previous version, **lines 80 - 83**:

“Scenarios of large-scale deployment of electrolytic hydrogen pose questions on the availability of a sufficient amount of resources for the required installation of renewable technologies and water electrolysis. Here, we focus on the demand and availability of land and water resources for prospective large-scale electrolytic hydrogen production using electricity from wind and solar generation.”

Revised version, **lines 75 - 84**:

“Large-scale deployment of electrolytic hydrogen raises concerns about the availability of sufficient land and water resources for the installation of solar photovoltaic panels, wind turbines and water electrolysis systems. Existing analyses ^{18,19,20,21} overlook or consider only global-scale assessments of land and water availability, independent of country-specific limits, when examining the production and trade of hydrogen from countries with abundant renewable resources to those with limited renewable energy potential. In this study, we focus on assessing the global demand and availability of land and water resources at the country level for prospective large-scale electrolytic hydrogen production using wind and solar electricity. By considering various land coverage scenarios for solar photovoltaics and wind turbines, we identify the countries where the local supply of electrolytic hydrogen could exacerbate land and water scarcity based on different levels of renewable technology penetration.”

5. The Journal’s standards for referencing must be considered.

We thank the reviewer for this remark. We have revised all references to comply with the journal formatting standard.

6. A list of Abbreviations must be given.

We provided a list of symbols, Greek letters and abbreviations at the end of the **Supplementary Information**.

7. To indicate the importance of hydrogen and its role in the future, the following papers are suggested to be considered:

"Hydrogen has found its way to become the fuel of the future." Future Energy 1.3 (2022): 11-12. <https://doi.org/10.55670/fpll.fuen.1.3.2>
"Hydrogen-A sustainable energy carrier." Progress in Natural Science: Materials International 27.1 (2017): 34-40.

We thank the reviewer for the suggestion. We added the references in the section **Methods – 5.1 Hydrogen demand**.

8. What are the limitations of low-carbon alternatives for hydrogen production, such as coupling of steam methane reforming with carbon capture and renewable natural gas from anaerobic digestion of biomass feedstock?

We thank the reviewer for this question. We realized that this section was lacking clarity and we modified it to provide information on the points raised more explicitly.

Previous version, **lines 67-69**:

“Constraints varying from the local availability of feedstocks (sustainable biomass feedstocks) [6], to the need of a new infrastructure for the capture, transport, and storage of carbon [10,11], and the management of leakage in the system for the production and transport of methane [12].”

Revised version, **lines 62-65**:

“However, these technologies face limitations, such as the local availability of sustainable biomass feedstocks ⁶, infrastructure requirements for carbon capture, transport, and storage ^{10,11}, methane leakage management ¹², and limited CO₂ capture rates at the plant level (55%-93%) ⁷.”

9. How can large-scale deployment of electrolytic hydrogen affect the availability of land and water resources for food production and other uses?

We thank the reviewer for the question.

To better clarify this point in the manuscript, we expanded the paragraphs in the **Introduction** section where we discuss the competition of land and water resources for food production and other uses.

Previous version, **lines 83 – 102**:

“Wind and solar energy infrastructure can have a significant direct impact on the environment in the areas where it is installed, in terms of habitat removal and biodiversity loss. This impact can extend beyond the land occupied, with indirect implications in the responses of animal species and ecosystems to the presence of infrastructure ¹⁸. Limitations in land availability are strongly influenced by the use of land for food production; increasing populations result in a progressive decline in the availability of arable land per capita ¹⁹. The rising production of biofuels for energy end-uses is one of the factors leading to an increase in land competition between energy and food systems ^{20,21}.”

Global demand for water has been increasing at 1% rate per year ²², in parallel with a progressive reduction in available water resources and increase in water pollution ²³. These trends have been driven by both population and economic growth. Unequal distribution of water availability combined with unequal economic growth have increased water scarcity and led to a competition in water use between energy, industrial and municipal systems ²⁴. Water demand for hydrogen production is projected to have relatively small hydrological implications at a global scale, much smaller than projected demand for water for food production ²⁵. However, local and regional water constraints have been recognized among the key factors that define the capability of a country to produce and export electrolytic hydrogen, together with the availability of renewable resources and infrastructure development ^{26,27}. Typically, sunnier regions are also drier and the majority of planned electrolyser capacity is projected to be located in regions facing water scarcity ^{28,29}.”

Revised manuscript, **lines 84 – 111:**

“Dedicating land to the installation of renewable technologies can reduce the availability of land for cropland expansion, hence limiting the amount of land dedicated to food production. The competition in land uses, combined with the rise in population, lead to a decrease in the per capita availability of arable land ²². The growing production of biofuels for energy also contributes to increased competition for land between energy and food systems ^{23,24}. Additionally, the infrastructure required for wind and photovoltaic energy can have significant direct environmental impacts, including habitat loss and biodiversity decline ²⁵. These impacts can extend beyond the occupied land area, affecting animal species and ecosystem responses ²⁶.

Global demand for water has been steadily increasing at a rate of 1% per year ²⁷, accompanied by a progressive decline in available water resources and an increase in water pollution ²⁸. These trends can be attributed to population growth, climate change, and economic development. The uneven distribution of water availability, coupled with varying economic growth rates among countries, has intensified water scarcity and led to competition in water usage among energy, industrial and municipal systems ²⁹. Total water demand for hydrogen production is projected to have relatively minimal hydrological implications on a global scale compared to water demand for food production ¹⁸. Among the end-uses, the estimated water demand for global electrolytic hydrogen in a net-zero emissions chemical industry is less than 1.5% of the water demand for food production ³⁰. However, local and regional water constraints have been recognized among the key factors that define a country’s capability to produce and export electrolytic hydrogen. At the country or local level, the concentration of plants manufacturing photovoltaic panels and wind turbines, or factories operating electrolyzers, in areas affected by water scarcity can exacerbate the limitations in water availability due to competing demands for water resources. These constraints are considered alongside the availability of renewable resources and infrastructure development ^{19,31}. It is worth noting that regions with abundant solar resources also tend to face water scarcity, and it is anticipated that the majority of planned electrolyzer capacity will be located in such regions ^{20,32}.”

10. Highlight the main drivers for the increase in hydrogen demand in the chemical sector from 2020 to 2050.

We modified the section describing the assumptions behind the hydrogen demand (**Methods – 5.1 Hydrogen demand**) by adding the following parts.

Revised version, **lines 535-536**:

“In our reference scenario, the global demand for ammonia in 2020 for chemical production amounts to 189 Mt/y, projected to reach 243 Mt/y by 2050⁴.”

Revised version, **lines 538-544**:

“The demand for fertilizers, historically driven by ammonia, continues to grow mainly in developing economies, while developed economies reach saturation. However, the production of fuel additives, explosives and synthetic materials (such as nylon, acrylic fibers and nitrile rubber) is expected to drive increased ammonia demand^{58,60}. In 2020, the global demand for methanol equals 100 Mt/y, projected to rise to 181 Mt/y for chemical production.”

Revised version, **lines 544-545**:

“The increased demand for methanol is primarily driven by its use as a fuel additive and as an intermediate product for HVCs, projected⁵⁸ to grow by 60% by 2050 compared to 2017. “

11. Highlight the hydrogen strategies in the countries that will experience the highest increase in per capita demand for hydrogen in the reference case for 2050.

We thank the reviewer for suggesting this addition.

We added the following text as section **S.1.3 Country-specific hydrogen strategies** of the **Supplementary Information**.

Revised version, **lines 47-95**:

S.1.3 Country-specific hydrogen strategies

In the following section, we examine the hydrogen strategies of four prominent countries that are projected to experience significant increases in hydrogen demand between 2020 and 2050.

China¹⁴

China primarily relies on coal gasification for hydrogen production, in contrast to other countries where natural gas is the main source. China’s initial hydrogen roadmap emphasized fuel cell vehicles and included purchase subsidies for light-, medium-, and heavy-duty vehicles. The country aimed to deploy over 33 000 fuel cell vehicles across five clusters of cities in a four-year demonstration project, which later increased to 50 000 vehicles by 2025. Recent plans involve producing 0.1 - 0.2 Mt/y of electrolytic hydrogen by 2025. The China Hydrogen Alliance predicts total hydrogen use to reach 130 Mt/y by 2060, accounting for 20% of China’s total final energy consumption.

Australia^{15,16}

Australia aims to become one of the top three hydrogen-exporting countries to Asian markets by 2030. As part of its net-zero goals, Australia plans to produce clean hydrogen at a cost below 2 \$/kg. The country has allocated 464 M\$ to establish clean hydrogen industrial hubs. The Australian national hydrogen strategy includes measures to support research and development of hydrogen supply chains and create favorable regulatory conditions.

France and other European countries ^{17,18,19,20,21,22}

The French hydrogen strategy is aligned with the other three major economies in the European Economic Area. France, Italy, Spain, and Germany plan the installation of 6.5 GW, 5 GW, 4 GW, and 5 GW of low-carbon hydrogen production plants by 2030, respectively. At the European scale, the RePowerEU Plan from the European Commission set a production target of 10 Mt/y by 2030, in addition to 10 Mt/y of hydrogen import. Among them, Belgium plans to base 100% of its supply on import due to the limited local renewable energy potential.

Japan ^{23,24}

On the supply side, Japan will develop international commercial-scale supply chains by 2030 to import 0.3 Mt/y of hydrogen. On the demand side, the Green Growth Strategy established the core targets of 3 Mt/y of hydrogen consumption by 2030 and 20 Mt/y by 2050. Additional actions include commercializing the use of hydrogen in the power sector, in energy-intensive industries, and in heavy-duty vehicles. Furthermore, the installation of stationary fuel cells for the off-grid supply of electricity and the development of liquefied hydrogen carrier vessels are part of the actions. The Japanese Green Growth Strategy has two additional targets: introducing 30% of hydrogen blending in gas-fired power plants and 20% ammonia-blending in coal-fired power plants to cover 1% of Japan's power generation. As a net importer of hydrogen, Japan has developed an international supply chain with Brunei and Australia.

United States ^{25,26,27}

The national hydrogen strategy of the United States builds on the Infrastructure Investment and Jobs Act (November 2021), the Hydrogen Energy Earthshot (June 2021) and the Inflation Reduction Act (August 2022). Scenarios have been developed indicating 10 Mt of hydrogen demand by 2030, 20 Mt/y by 2040 and 50 Mt/y by 2050. The strategy builds around three targets: (i) clean hydrogen use in the highest-value applications, (ii) reduction of the cost of clean hydrogen production, and (iii) creation of clean hydrogen hubs for large-scale production. The strategy aims at lowering the cost of hydrogen production to 2 \$/kg by 2025 and 1 \$/kg by 2030. More recently, the Inflation Reduction Act (IRA) has created a wide array of energy tax incentives providing a production tax credit of up to 3 \$/kg for low-carbon hydrogen covering the first 10 years of operation of a production plant.

12. Highlight the main regions affected by water scarcity in hydrogen production, and how do they compare to regions with no water scarcity?

The concern of the reviewer regarding the main regions affected by water scarcity is addressed in section **2.5 Water Scarcity**. In this section of the manuscript, we compute the impact of hydrogen production on the existing water scarcity. From the revised manuscript, **lines 293 – 299**: “[...] in 37 countries already facing water scarcity, hydrogen production exacerbates water scarcity. The most affected countries are in the North African and Middle

East regions (MENA). Our analysis reveals that 5 countries experience an increase in water scarcity greater than 5% when using solar panels, while 16 countries experience a relatively minor exacerbation below 1% (Figure 5 (a)). Specifically, Oman and Qatar demonstrate an increase in water scarcity greater than 5% when using solar panels (Figure 5 (a)), and Qatar experiences this increase when utilizing wind technologies (Figure 5 (b)).”

Additionally, in the revised manuscript we included a section focusing on the impact of water requirements for hydrogen production in countries where water scarcity is not present, and is not created by hydrogen demand: **Supplementary Information - Section S.4.2 Impact of water requirements for hydrogen production.**

Revised manuscript, lines 213 – 231 of **Supplementary Information:**

S.4.2 Impact of water requirements for hydrogen production

Figure S.3 illustrates the additional water requirement in countries that are not affected by water scarcity, compared to the scenario without hydrogen production (see Methods - Section 5.4.2 Water scarcity - Equations 9, 10). Countries such as Russia, European nations, and Southern African countries like Namibia and Angola exhibit significant increases in water demand. In these countries, the water requirement for hydrogen production represents more than 4% of the total water withdrawals for agriculture, industry, and municipal purposes. Figure S.3 complements Figure 5 in Main text - Results which illustrates the impact of water for hydrogen production in countries where water scarcity arises.

Figure S.3: Relative increase of water use due to additional water requirements for hydrogen production in 2050 through water electrolysis, compared to current water withdrawals for agricultural, industrial and municipal activities. Power production from (a) solar, (b) hybrid wind and solar and (c) onshore wind. Countries depicted without color indicate areas that experience water scarcity regardless of hydrogen demand. Figure 5 illustrates the impact of water demand for hydrogen production in countries where water scarcity is present. It is worth noting that the demand for water in hydrogen production alone does not lead to water scarcity where it is not already present. The gray-colored countries indicate areas for which data is unavailable.

13. How does the fraction of eligible land coverage assumed affect the number of countries limited by land and water resources in hydrogen production?

In the revised manuscript we included a section focusing on the impact of the fraction of eligible land coverage on the number of countries limited by land and water resources: **Supplementary Information - S.9 Sensitivity number of countries with land and water scarcity.**

Revised manuscript, lines 315 – 332 of **Supplementary Information:**

“

S.9 Sensitivity number of countries with land and water scarcity

The extent of land and water scarcity affecting countries for power production from solar panels, hybrid solar and wind systems, and wind turbines depends on the assumed parameter of eligible land coverage (f coverage). Figure S.11 illustrates the varying number of countries affected across a wide range of f coverage values, ranging from 1% to 95%. The number of countries with concurrent presence of land and water scarcity is 14 for solar panel power production and 28 for wind turbines, when assuming only 1% of eligible land coverage. As the value of coverage increases, the concurrent presence of land and water scarcity decreases significantly. Hybrid wind and solar systems present an intermediate case in terms of both water and land use. Due to the lower water requirements of power production from wind turbines, the number of countries affected by both land and water scarcity slightly decreases compared to the case of solar panels.

Figure S.11: Number of countries affected by concurrent land and water scarcity for power production from (a) solar panels, (b) hybrid solar and wind systems, and (c) wind turbines. The figure illustrates the sensitivity of the number of countries experiencing both land and water scarcity based on the eligible land coverage parameter.

“

The concern of the reviewer is also addressed in **Section 3 – Discussion**.

Revised manuscript, **lines 318 – 345**:

“

Figure 6 combines the analysis of land and water scarcity, highlighting countries constrained by both land and water resources (red), land resources alone (yellow), water resources alone (blue), and those with no land and water limitations (green). These findings are highly dependent on the energy generation technology and the assumed fraction of eligible land coverage (f coverage) (see sensitivity analysis in Suppl. Inf. - Section S.9 Sensitivity number of countries with land and water scarcity). Water scarcity is predominantly observed in the MENA region, South Africa, India, China, and certain countries in Central Asia. Among the countries facing both land and water scarcity (red countries in Figure 6 (a,b)), Trinidad and Tobago experiences it when considering 100% eligible land (f coverage = 100%) and full water resource exploitation in the case of solar panels, while Belgium and South Korea experience it in the case of wind power. The comparison of results for various demand scenarios is summarized in Suppl. Inf. - Section S.9 Sensitivity number of countries with land and water scarcity.

In contrast, when considering only 5% of eligible land for the installation of solar panels (f coverage = 5%) and hydrogen demand from our reference cases, Belgium, Trinidad and Tobago, South Korea, and Dominican Republic are the countries that experience both land and water scarcity (red countries in Figure 6 (e)). However, if we assume renewable electricity solely generated from onshore wind, the number of countries facing both land and water scarcity increases to 20, including Spain among the European countries, South Africa, Middle East countries, India and China (red countries in Figure 6 (f)). Land scarcity alone emerges in European countries under more restrictive scenarios of eligible land coverage with renewable technologies. The disparity between using solar and wind energy is substantial. With 10% eligible land coverage (f coverage = 10%), three countries experience contemporary land and water scarcity for solar (Figure 6 (c)) while the number rises to 14 countries for wind (Figure 6 (d)). Under the assumption of 5% eligible land coverage (f coverage = 5%), 45 and 94 countries encounter land or water scarcity, or both, when utilizing solar (Figure 6 (e)) and wind energy (Figure 6 (f)), respectively. It should be noted that land scarcity in countries with large geographical extensions, such as the United States, Brazil, Russia and China, arises due to a significant portion of land being covered by forests and/or agriculture. Countries that experience no scarcity under any condition include Australia, Canada, Argentina, Mongolia, Namibia, and a substantial number of countries in Central Africa.

Figure 6: Land and water scarcity induced by projected hydrogen production in 2050 for all countries worldwide, considering various fractions of land coverage for (a,c,e) solar and (b,d,f) onshore wind power production. The color-coded regions represent the scarcity status: green for no scarcity, red for both land and water scarcity, yellow for land scarcity only, and blue for water scarcity only. Gray-colored countries indicate unavailable data.

”

14. Explain the potential consequences for countries limited by both land and water resources in hydrogen production, and what alternatives do they have to achieve net-zero targets?

The presence of land scarcity (**Methods – 5.4.1 Land scarcity**) is caused by a limited availability of land which can be dedicated to power production by feeding electrolyzers without the creation of conflicts with other land end-uses, i.e., agriculture, urban areas and forests. The presence of water scarcity (**Methods – 5.4.2 Water scarcity**) is caused by a limited availability of water which can be dedicated to hydrogen production by feeding electrolyzers without the creation of conflicts with other water end-uses, i.e., agriculture, municipal, industrial and environmental flow requirements.

In the case of electrolytic hydrogen considered in our study, the first potential consequence for countries limited by land and water scarcity is a higher reliance on the import of electricity or hydrogen. A higher import of electricity can be achieved through the expansion of the high-voltage transmission grid across countries. The import of hydrogen would require new pipelines or the establishment of new maritime trading routes. We discuss this potential consequence in the third paragraph of **Section 3 – Discussion**, with respect to countries having a contemporary presence of land and water scarcity.

The second potential consequence is a limitation in the demand for hydrogen. For the chemical sector in particular, existing plants rely on conversion processes fed with local production of hydrogen from fossil resources. These plants might have to relocate to regions where land and water scarcity don't represent a limitation in the local production of low-carbon hydrogen (offshoring). Considering the substantial contribution that large-scale industrial plants have to the economy of a country, the offshoring of multiple production plants could determine a deindustrialization of the country. We discuss the scenario of offshoring in the fourth paragraph of **Section 3 – Discussion**, where we highlight the uncertainty in the future location of the hydrogen production and consumption

points. We emphasize that access to cheap renewable energy is a main driver of offshoring of production plants in the energy-intensive industry.

A third consequence concerns the dependency on traditional technologies producing low-carbon electricity. Namely, electricity from nuclear power plants or from conventional power plants which rely on fossil resources, coupled with carbon capture technologies. In this case, while the impact on land demand would be limited by the higher energy production per unit of land, water scarcity would still represent a limitation to their use.

REVIEW #3

Remarks to the author

This article provides excellent insight on often overlooked considerations in the implementation of a global hydrogen economy, particularly related to land and water scarcity as far back in the process as the production of renewable electricity by solar and wind technologies. While the article agrees with conclusions that are widely accepted in the scientific community that Earth provides an abundance of water needed to meet the stoichiometric needs of water electrolysis, this study considers the very important contributions of solar panel and wind turbine production. The country-by-country breakdown of green hydrogen production capabilities is critical to understanding how and where the hydrogen economy will develop and change the landscape of the global economy. My recommendations to the authors are related to the assumptions and approaches used in these projections, but with the revisions below the article in my opinion is thus appropriate for publication in Nature Communications.

We would like to express our gratitude to the reviewer for the appreciation of our work. We have carefully considered the recommendations and made the necessary revisions to enhance the clarity of the manuscript. We hope that the updated work meets the required level of clarity and addresses any concerns raised.

1. Some grammar corrections for run-on sentences or other incorrect sentence structure, including:
 - “Constraints varying from the local... transport of methane” (Lines 67-69)
 - “Despite... water resources.” (Lines 393-395). Consider using “Although” as an alternative to “Despite”
 - “With the exception of Trinidad... municipal demand) (Lines 227 – 231). The sentence structure here sounds awkward.

We thank the reviewer for highlighting the errors. We modified the sentences as follows.

Previous version, **lines 67-69**:

“Constraints varying from the local availability of feedstocks (sustainable biomass feedstocks) [6], to the need of a new infrastructure for the capture, transport and storage of carbon [10,11], and the management of leakage in the system for the production and transport of methane [12].”

Revised version, **lines 62-65**:

“However, these technologies face limitations, such as the local availability of sustainable biomass feedstocks 6, infrastructure requirements for carbon capture, transport, and storage ^{10,11}, methane leakage management ¹², and limited CO₂ capture rates at the plant level (55%-93%) ⁷.”

Previous version, **lines 393-395**

“Despite water consumption for hydrogen production is not expected to create water scarcity in countries where it does not already exist (except in Trinidad and Tobago), it can lead to a further exacerbation of unsustainable use of water resources.”

Revised version, **lines 479-487**:

“While water requirements for hydrogen production is not expected to create water scarcity in countries where it does not already exist (except in Trinidad and Tobago), it does have the potential to exacerbate the unsustainable utilization of water resources.”

Previous version, **lines 227-231**:

“With the exception of Trinidad and Tobago in case of hydrogen production from solar panels, in our reference cases, the demand of water for hydrogen production does not create water scarcity anywhere in the world when water scarcity is not already present, being water withdrawals for hydrogen production negligible compared to the sum of the water demand in other sectors (agriculture, industry and municipal demand) (Suppl. Inf. S.4).”

Revised version, **lines 287-293**:

“Except for Trinidad and Tobago, where hydrogen production from solar panels can lead to water scarcity, the assumed reference scenario for hydrogen demand does not create water scarcity anywhere in the world if water scarcity is not already present (the relative increase in water withdrawals due to hydrogen production is shown in Suppl. Inf. – Section S.4.2 Impact of water requirements for hydrogen production). This is because water withdrawals for hydrogen production are negligible compared to total water withdrawals in other sectors (agriculture, industry, and municipal demand) (see Suppl. Inf. - S.4 Water demand).”

2. Some suggestions for clarity in figures: Figure 1: Is the left-hand side y-axis in kW_{el} per person? If so, could you make this unit label agree with Figure 2 (listed as per capita)

We thank the reviewer for spotting the inconsistency between the two figures. We modified the y-axis label from “capacity electrolytic hydrogen (kW_{el}/p)” to “electrolytic hydrogen capacity (kW_{el} per capita)”.

3. In lines 307-310, the authors indicate that the current hydrogen demand is used in the projections for future hydrogen demand in the chemical and petrochemical industries. What factors contribute to the uncertainty and make projections difficult, compared to the projections that are used for other sectors? If so, please reconcile these industries when comparing to other sectors that have undergone projection in the methods (such as ammonia and methanol production projections referenced in lines 460-461).

We thank the reviewer for the request of clarification.

The same projection method has been applied to all the sectors. In **Methods – Section 5.1** we provide details on the steps involved in the derivation of the hydrogen demand per sector, schematically summarized in **Supplementary Information – S.1.1 Hydrogen Demand – Figure S.1**. We quantified the hydrogen demand for the following sectors: ammonia production, methanol production, cement production, refineries, transportation, light industry and steel production. For each sector, our analysis relies on data from 2019 and derived projections for 2050 based on the variation of population:

- **lines 562-564** for methanol and ammonia production
- **lines 579-581** for cement production

- **lines 603-605** for refineries, transportation, light industry and steel production

In the **Discussion** section of the previous version beginning with **lines 307-310**, we wanted to emphasize that the current location of the production plants of chemical products (such as ammonia and methanol) can vary substantially by 2050. Hence, relying our projections for 2050 on country-specific data of energy consumption in 2020 can be a substantial source of error. Among the reasons in the variation of the location of the production plants stems the lower cost of energy which could drive a relocation by 2050. To avoid this error, an alternative approach to predict the hydrogen demand for the chemical sector in 2050 might have been using the demand of final products in 2050. However, this demand is affected by country-specific economic trends and market dynamics which would require tailored modelling. The dependence of the future location of the plants from the cost of energy applies to all the energy-intensive sectors such as steel, cement, ammonia and methanol. We modified the text to underline this aspect.

Previous version, **lines 307-312**:

“For the chemical and petro-chemical industry, we decided to locate hydrogen demand based on existing production plants, instead of the potential future location of consumption of chemicals. The future location of both the production and consumption of chemicals is subject to uncertainty and dependent on multiple factors. In energy-intensive industries, like steel, chemicals and petrochemicals, the cost of energy represents a major component in the cost breakdown of final products [28]. [...]”

Revised version, **lines 370-384**:

“When considering energy-intensive industries such as steel, cement, ammonia and methanol, we made the decision to allocate hydrogen demand based on the existing production plants rather than the potential future locations of final products consumption. This approach is due to the uncertainties surrounding the future locations, which depend on various factors. For energy-intensive industries, the cost of energy is a significant component in the overall cost of final products²⁰. Historically, production plant locations for these industries have been driven by access to inexpensive fossil resources⁴¹. However, the implementation of carbon emissions reduction initiatives has encouraged the adoption of direct or indirect electrification methods, including electrolytic hydrogen, to replace fossil-based processes. Access to affordable local renewable energy has become a crucial factor influencing the relocation of these industries. Other factors contributing to relocation decisions include material, labor, and capital costs, which are influenced by country-specific policies and corporate strategies. Future consumption patterns in the chemical and petrochemical industry are influenced by market dynamics and the adoption of new end-use technologies. Due to the numerous factors that can affect these industries, we are cautious about making assumptions regarding their displacement. Therefore, our analysis in this study is based on historical data from 2020.”

4. In line 416, the authors indicate that they corrected data from 2019 to meet the need for 2020 data excluding the impact of COVID-19. Is it feasible to just 2019 data directly as the earlier case study to avoid further approximation?

We thank the reviewer for the observation. The rationale behind the calculation of the hydrogen demand has been to consider scenarios of total hydrogen demand at the global

scale from a widely-used report (IEA, 2021. Net Zero by 2050) ⁴. These scenarios extend from 2020. Thus, we would either need to bring the 2020 global numbers of hydrogen demand from the report ⁴ to the 2019 World Energy Balances data ⁵⁵ or vice versa. We chose to align our analysis with the years of hydrogen demand presented by the (IEA, 2021. Net Zero by 2050) ⁴.

To avoid the impact of COVID-19 on our baseline reference year, two approaches are possible:

Approach 1 – 2020 scenario:

- a. using the scenario of global hydrogen demand for 2020, provided by (IEA, 2021. Net Zero by 2050) ⁴,
- b. correcting country level World Energy Balances data ⁵⁵ from 2019 to derive values for 2020 unaffected by COVID-19,
- c. disaggregating the global hydrogen demand for 2020 based on corrected country level data for 2020

Approach 2 – 2019 scenario:

- a. correcting the scenario of global hydrogen demand from 2020 data to 2019,
- b. using country level World Energy Balances data ⁵⁵ in 2019 to disaggregate the global demand,
- c. disaggregating the corrected global hydrogen demand in 2019 to country level based on country level data in 2019

Considering that the disaggregation for 2050 is based on country level projected data for 2050, we preferred to adopt Approach 1. In this way, we remained consistent with the scenario of global hydrogen demand in 2020 from the reference source. Following the comment of the reviewer, we slightly modified the text.

Previous version, **lines 415-419**:

“To exclude demand variations directly correlated with the impact of COVID-19, data for 2020 are derived from 2019 with corrections only based on the variation of population.”

Revised version, **lines 503-505**:

“To exclude demand variations directly correlated with the impact of COVID-19, country-specific data for 2020 are derived from country-specific data for 2019 with corrections only based on the variation of population.”

5. In section 5.3.1, the authors describe how they determine the energy production yield of solar and wind energy, and in sections 5.3.2 and 5.3.3 they describe the land and water requirements for each of these energy production methods. However, it seems that the cases are divided into countries using 100% solar energy or 100% wind energy for the data in the manuscript. Could the authors consider cases that include a combination of wind and solar resources inside one country?

We thank the reviewer for the comment. In addition to the cases of 100% solar and 100% wind energy, we took into consideration the case of hybrid production composed by 60% of eligible land covered with wind turbines and 40% of eligible land covered with solar panels. The weights used for the combination of solar and wind are based on (Heide et al., 2010).

In the **Supplementary Information** of the revised version of the manuscript, we added section **S.5 Combination of solar and wind**, presenting the results specific for this combination of technologies.

Revised manuscript, **lines 232 – 294**:

“

S.5 Combination of solar and wind

The energy production per unit area from hybrid solar photovoltaics and wind turbine systems is computed as:

$$S_i^{wind + solar} = 0.6 S_i^{wind} + 0.4 S_i^{solar} \quad \forall i \in N \quad (11)$$

with:

- $S_i^{wind + solar}$ (TWh/km²/y) yearly energy production from hybrid wind and solar systems in cell i .

Here, we choose hybrid systems with a production composed of 60% wind turbines and 40% solar panels, corresponding to the minimization of the storage capacity needed 49. In the following, we show the list of figures presented in the article for the case of energy production per unit area from hybrid solar photovoltaics and wind turbine systems.

In the following, we present the corresponding figures for the case of hybrid solar and wind systems, which are equivalent to those presented in the main text.

Figure S.4: Per capita land requirements to meet the hydrogen demand in 2020 and 2050 based on hybrid solar and wind systems. The interval bars represent the range of land required in 2050 based on different reference and extreme scenarios of global hydrogen demand in 2050: 92 Mt/y (smallest), 400 Mt/y (reference), 646 Mt/t (largest). The gray areas (and pink areas) indicate fractions (and multiples) of the eligible land, defined in Methods - Section 5.4.1 Land scarcity as the 100% threshold, for comparison with the land requirements for hydrogen demand in each country. Different energy generation technologies require varying amounts of land for the same amount of hydrogen demand. Countries are selected based on the highest total demand of hydrogen in 2050 and are ordered according to the amount of eligible land for solar panels (excluding Trinidad and Tobago for scaling reasons).

Figure S.5: Water withdrawals per capita to satisfy hydrogen demand in 2020 and 2050, based on hybrid solar and wind systems. The gray areas represent the fractions of available water resources (water resources minus environmental flow requirements) in each country, corresponding to the water withdrawals. The red areas indicate water withdrawals exceeding the water availability limit (100%, 250%, and 500%). The purple, blue and yellow stacks represent water withdrawal in agriculture, industry and municipality, respectively. The red stack represents the water required for hydrogen demand in 2020 and 2050. Total water withdrawals exceeding 100% indicate the presence of water scarcity. The countries were selected from the top 30 in terms of total hydrogen demand in 2050 and are ordered according based on the amount of available water.

Figure S.6: Land scarcity induced by hydrogen production in 2050 across worldwide countries, considering various fractions of land coverage (a, b, c) for hybrid solar and wind systems. The degree of land scarcity is closely tied to the assumed eligible land coverage (f coverage) for renewable technologies in each country, reflecting economic and socio-political constraints. Countries depicted in gray lacked available data.

Figure S.7: Exacerbation of water scarcity resulting from additional water requirements for hydrogen production in 2050 through water electrolysis, compared to current water withdrawals for agricultural, industrial and municipal activities. The demand for water in hydrogen production alone does not create water scarcity in countries where it is not already present. However, the additional water demand for hydrogen production can exacerbate scarcity in countries already affected by water scarcity. Countries without color in the figure do not experience water scarcity. Gray-colored countries indicate unavailable data.

Figure S.8: Land and water scarcity induced by projected hydrogen production in 2050 for all countries worldwide, considering various fractions of land coverage (a, b, c) for hybrid solar and wind systems. The color-coded regions represent the scarcity status: green for no scarcity, red for both land and water scarcity, yellow for land scarcity only, and blue for water scarcity only. Gray-colored countries indicate unavailable data.

6. The authors have decided to omit the power/energy storage sector from their hydrogen demand projections due to uncertainties in the amount of hydrogen allocated for this use in 2050. However, as the authors indicate, this could lead to an increase in the hydrogen demand of 24% by 2050. If there is interest in including this sector, the following sources provide information about projections of the usage of renewables for power in 2050 and the need for about 10-20% of storage as hydrogen.
- U.S. Energy Information Administration, International Energy Outlook 2020 (IEO2020), 2020.
 - M. Sterner, I. Stadler, Handbook of Energy Storage Demand, Technologies, Integration, 2nd ed., Springer, 2019.
 - I. Petkov, P. Gabrielli, Power-to-hydrogen as seasonal energy storage: an uncertainty analysis for optimal design of low-carbon multi-energy systems, Appl. Energy. 274 (2020) 115197. <https://doi.org/10.1016/j.apenergy.2020.115197>.

- d. A.O. Converse, Seasonal energy storage in a renewable energy system, in: Proc. IEEE, 2012: pp. 401–409. <https://doi.org/10.1109/JPROC.2011.2105231>.

We warmly thank the reviewer for this observation and for providing useful references in support of the addition of the hydrogen demand related to the power/energy storage sector.

We agree with the reviewer on the possibility of finding estimates of the future demand of hydrogen in the power sector based on the penetration of renewables. However, in the majority of the cases these estimates refer to a geographical coverage of multiple countries (i.e., global aggregation, aggregation based on climate regions, continental aggregation, ...).

Some works study the penetration of hydrogen by using a simplified energy system, coherent with the objective of the study, but with limitations on the possibility of generalizing the results to a more realistic estimation of country-specific demand of hydrogen in the power sector in 2050. The table summarizes and comments the information about the projections of hydrogen in the power sector from the reference studies considered.

source	value	scale demand	comments
IRENA (2022), Global hydrogen trade to meet the 1.5°C climate goal: Part I – Trade outlook for 2050 and way forward, International Renewable Energy Agency, Abu Dhabi.	Fraction of hydrogen in the power sector: ~ 28 % of total hydrogen demand	Global demand	- consideration of demand in the power sector will be based on integrated gas and power models - share of power consumption excluded from the analysis
U.S. Energy Information Administration, International Energy Outlook 2020 (IEO2020), 2020.	Fraction of renewable generation in 2050: 38%	United States	- valid reference for global penetration of renewables - substantial assumptions would be needed to derive country-specific values of renewable penetration
I.Petkov, P. Gabrielli, Power-to-hydrogen as seasonal energy storage: an uncertainty analysis for optimal design of low-carbon multi-energy systems, Appl. Energy. 274 (2020)	Fraction of hydrogen storage: - 3-5% of annual net electric demand in warm climate zones - 7-8% of annual net electric demand in cold climate zones	4 climate zones based on aggregation of European countries	- hydrogen storage for power production only competes with batteries. Other technologies for seasonal storage, like hydropower, are not included - the energy system in this analysis only considers fuel cells for power production from hydrogen. Other technologies, like 100% fueled gas turbines are not considered
A.O. Converse, Seasonal energy storage in a renewable energy system, in: Proc. IEEE, 2012: pp. 401–409.	Storage capacity as percentage of annual electricity produced: 4.9% - 31.7%	Method that can be applied at country-scale. Data requirements: monthly data of electrical energy consumption and generation from solar, wind and hydro	- this is a valid method for an estimation of the required seasonal storage based on monthly data of renewable generation and electrical energy consumption - electrical energy consumption and hydro generation for 2050 could be derived from 2020 data, although they would require the creation of a synthetic distribution over the 12 months of the year - renewable production from solar and wind in 2050 would require country-specific assumptions on the penetration of wind and solar for each country in the world. One method of calculation would require considering the amount of electricity produced from nuclear power plants or imported in each country. Another method would require country-specific assumptions on the fraction of land covered by solar and wind. However, this method would be inconsistent with the analysis of scarcity which assumes fractions of land coverage, representative of highly uncertain economic and socio-political constraints.

To the best of our knowledge, we did not find country-specific projections of the hydrogen demand in the power sector for 2050. The method proposed in (Converse, 2012) is valid to identify the fraction of seasonal storage capacity in a country, however it would require a quantification of the penetration of solar and wind production in the energy mix in 2050. Even assuming a top-down approach to disaggregate the global demand for hydrogen to a country level, we were not able to identify a metric, applicable to all the countries, on which we could base the disaggregation. Under the assumption of a full conversion of gas turbines fueled with natural gas to 100% hydrogen-fueled, the capacity installed could be used as a metric for a country level disaggregation of the regional/global estimation of the hydrogen demand in the power sector. However, part of the hydrogen stored might be used in fuel cells yet to be installed. The installation of gas turbines in the energy system might also drastically change over the time horizon of 25/30 years.

We believe that a sufficiently accurate estimate of the country-specific amount of hydrogen in the power/energy storage sector for 2050 requires the combined modelling of the power and the gas sector with a sufficiently accurate representation of the country-specific energy mix in 2050 (including, e.g., the penetration of nuclear power and import of electricity). The model should include country-specific maximum storage capabilities and include a competition among different storage technologies (i.e., hydropower, hydrogen and derived electrofuels, batteries) whose availability depends on specific countries.

Alternatively, we considered the possibility of assuming a fixed fraction of the total hydrogen demand for the power sector in any country in the world. However, this approach would have been equivalent to assume a scenario of higher total hydrogen demand. This case is included in the additional sensitivity analysis on the total demand of hydrogen presented in section **S.9 Sensitivity number of countries with land and water scarcity**, based on a scenario of low (92 Mt/y) and high hydrogen demand (646 Mt/y).

In conclusion, we decided to omit the hydrogen demand for the power sector.

REVIEWERS' COMMENTS

Reviewer #1 (Remarks to the Author):

I am glad to share with the authors that I am fine with the replies to all the comments I provided to the first version of the manuscript. All comments have been adequately addressed, either through amendments in the text/figures, or justifying why they were not, or fully implemented. Therefore, I agree with the publication of the manuscript.

Reviewer #2 (Remarks to the Author):

Accept

Reviewer #3 (Remarks to the Author):

The authors have fully and adequately addressed my comments. I particularly appreciate the addition of a wind + solar case study that was thoroughly represented in the SI. I recommend for publication in Nature Communications. Excellent work!

Manuscript No.: NCOMMS-23-12575A

We would like to extend our sincere gratitude to the reviewers for their constructive suggestions, which have significantly enhanced the quality of the article.

The positive reception of the proposed revised version by all the three referees brings us great satisfaction and reinforces our belief in the merit of the improvements proposed.

REVIEW #1

I am glad to share with the authors that I am fine with the replies to all the comments I provided to the first version of the manuscript. All comments have been adequately addressed, either through amendments in the text/figures, or justifying why they were not, or fully implemented. Therefore, I agree with the publication of the manuscript.

REVIEW #2

Accept.

REVIEW #3

The authors have fully and adequately addressed my comments. I particularly appreciate the addition of a wind + solar case study that was thoroughly represented in the SI. I recommend for publication in Nature Communications. Excellent work!